 eLife

# Area-specific development of distinct projection neuron subclasses is regulated by postnatal epigenetic modifications

**Kawssar Harb[1,2,3], Elia Magrinelli[1,2,3], Céline S Nicolas[1,2,3], Nikita Lukianets[1,2,3], Laura Frangeul[4], Mariel Pietri[1,2,3], Tao Sun[5], Guillaume Sandoz[1,2,3], Franck Grammont[1,6], Denis Jabaudon[4], Michèle Studer[1,2,3*†], Christian Alfano[1,2,3*†]**

[1]Institut de Biologie Valrose, University of Nice Sophia Antipolis, Nice, France; [2]Institut de Biologie Valrose, Institut national de la santé et de la recherche médicale, Nice, France; [3]Centre national de la recherche scientifique, Institut de Biologie Valrose, Nice, France; [4]Department of Basic Neurosciences, University of Geneva, Geneva, Switzerland; [5]Department of Cell and Developmental Biology, Weill Medical College of Cornell University, New York, United States; [6]Laboratoire J.A. Dieudonné, Nice, France

**\*For correspondence:** michele.
studer@unice.fr (MS); Christian.
ALFANO@unice.fr (CA)

[†]These authors contributed
equally to this work

**Competing interests:** The
authors declare that no
competing interests exist.

**Reviewing editor:** Jonathan A
Cooper, Fred Hutchinson Cancer
Research Center, United States

**Abstract** During cortical development, the identity of major classes of long-distance projection neurons is established by the expression of molecular determinants, which become gradually restricted and mutually exclusive. However, the mechanisms by which projection neurons acquire their final properties during postnatal stages are still poorly understood. In this study, we show that the number of neurons co-expressing Ctip2 and Satb2, respectively involved in the early specification of subcerebral and callosal projection neurons, progressively increases after birth in the somatosensory cortex. Ctip2/Satb2 postnatal co-localization defines two distinct neuronal subclasses projecting either to the contralateral cortex or to the brainstem suggesting that Ctip2/Satb2 co-expression may refine their properties rather than determine their identity. Gain- and loss-of-function approaches reveal that the transcriptional adaptor Lmo4 drives this maturation program through modulation of epigenetic mechanisms in a time- and area-specific manner, thereby indicating that a previously unknown genetic program postnatally promotes the acquisition of final subtype-specific features.

## Introduction

The mammalian cerebral cortex is tangentially subdivided into several functional areas allowing effective interactions with the external world by organizing sensory information into a coherent perceptual model of the environment. All neocortical areas are radially organized into six neuronal layers that show prominent diversities in features and functions despite similarities in their laminar organization. This is mainly due to differences in the molecular identity, morphology and long-range connectivity of residing neurons (*Greig et al., 2013*; *Huang, 2014*; *O'Leary et al., 2007*). Cortical projection neurons (PNs) can be subdivided into three major classes: (i) the intra-telencephalic (IT) neurons projecting to ipsilateral and/or, through the corpus callosum, to contralateral regions of the telencephalon (i.e. callosal projection neurons – CPN); (ii) the cortico-thalamic (CT) neurons projecting to different dorsal thalamic nuclei; and (iii) the pyramidal-tract (PT) neurons, also defined as subcerebral projection neurons (SCPN), that innervate different subcerebral targets, such as the striatum, the brainstem, and the spinal cord (*Harris and Shepherd, 2015*; *Shepherd, 2013*).

**eLife digest** The cerebral cortex is part of the outer layer of the mammalian brain, and it is important for a range of processes, including sensing, movement and conscious thought. The cerebral cortex is subdivided into several areas that are deputed to different functions. Each area is composed of an astounding variety of cells called projection neurons, which send information from the cerebral cortex to distant parts of the brain.

There are three main types of projection neurons, which each connect to a different brain region. However, when projection neurons first form in the embryo, they are all broadly similar. They then activate a combination of genes that determine their identity and behaviour through the activity of a vast range of transcription factors (proteins that control gene expression). At first, most of these transcription factors are active in more than one type of cortical neuron, but after the animal is born, these proteins become increasingly restricted to just one type of neuron. In this way, the major classes of projection neurons are specified. However, the mechanisms defining the remarkable variety of projection neuron subtypes in the cerebral cortex are still largely unknown.

Two of the transcription factors that act in the development of the major classes of projection neurons are called Satb2 and Ctip2. Satb2 prevents the activity of Ctip2, since the two proteins have opposite effects. However, some neurons in newborn animals produce both of these transcription factors.

Using mouse models, Harb et al. found that just after birth the number of projection neurons that express both Ctip2 and Satb2 increases in the cortical area that processes touch sense information – called the somatosensory cortex. These neurons are divided into two subclasses, each of which communicates with a different part of the brain. This suggests that Ctip2/Satb2 co-expression defines two subgroups of the major projection neuron classes rather than specifying new cell types.

Further investigations revealed that after birth Satb2 and Ctip2 are co-expressed in neurons due to another protein (called Lmo4) that modifies the structure of the DNA region that contains the Ctip2 gene. This prevents Satb2 from repressing Ctip2.

This work demonstrates that the great variety of projection neurons in the mammalian cerebral cortex is not due to the existence of several genetic programs directing the development of each single neuronal subtype. Instead, this variety is due to mechanisms that modify and refine, after birth, the processes that specify the major projection neuron classes. The main challenge in the future will be deciphering all the mechanisms that tilt the balance toward a given neuron subtype, and investigating whether and how this balance can be altered.

Major molecular determinants of CPN and SCPN neuronal classes (*Greig et al., 2013*; *Leone et al., 2008*; *Molyneaux et al., 2007*; *Srinivasan et al., 2012*) act by both promoting the identity of a given neuronal class and repressing alternative ones (*Fame et al., 2011*; *Greig et al., 2013*; *Kiritani et al., 2012*; *Molnar and Cheung, 2006*; *Molyneaux et al., 2007*; *Reiner et al., 2010*; *Sohur et al., 2014*). For example, Satb2, a chromatin remodeling protein, drives CPN specification and axonal pathfinding, and represses the expression of the transcription factor Ctip2 (gene name *Bcl11b*), which controls the connectivity of SCPN (*Alcamo et al., 2008*; *Arlotta et al., 2005*; *Baranek et al., 2012*; *Britanova et al., 2008*; *Srivatsa et al., 2014*). However, Satb2 was shown to control also subcerebral connectivity (*Leone et al., 2014*), indicating that the final acquisition of a given cell type is not based on the function of a unique transcriptional regulator but most probably on the combination of more determinants expressed at different levels. This is in agreement with observations performed on early embryonic stages, where neuronal types are not yet fully specified and factors with mutually exclusive functions largely overlap. Notably, expression of transcriptional determinants tends to segregate after birth in a cell- or time-specific manner when neurons differentiate and the major neocortical classes become more distinct from each other (reviewed in *Greig et al., 2013*). Nevertheless, it is still not clear whether the transcriptional regulators acting during early stages of neuronal specification play any additional roles during postnatal maturation when PNs acquire their final properties. Even if antithetic factors promoting callosal or subcerebral PNs can co-localize in a small subset of neurons after birth (*Azim et al., 2009*; *Baranek et al., 2012*;

*Leone et al., 2008*; *Tomassy et al., 2010*), it is not known whether their co-expression has a functional meaning, nor whether this hybrid molecular population corresponds to a permanent subgroup of cortical PNs. In particular, the persistence of few double Ctip2/Satb2-positive neurons at early postnatal stages of corticogenesis represents a conundrum, since Satb2 is a strong repressor of Ctip2 (*Alcamo et al., 2008*; *Baranek et al., 2012*; *Britanova et al., 2008*; *Leone et al., 2014*). Moreover, Satb2 and Ctip2 regulate the expression of two netrin1 receptors (Unc5C and DCC, respectively) with opposite effects on axon guidance (*Srivatsa et al., 2014*).

Here, we show that Ctip2/Satb2 co-expression does not delineate a transient stage of cortical development but instead distinct subpopulations of major PN classes, whose number progressively increases in the postnatal somatosensory mouse cortex. Double Ctip2/Satb2 (C/S+)-expressing cells define at least two neuronal subclasses, which project either to the brainstem or to the contralateral cortex and retain unique molecular, morphological, and electrophysiological features that distinguish them from single Ctip2- or Satb2-expressing cells. Moreover, we demonstrate that the transcriptional adaptor Lmo4 allows the co-localization of Ctip2 and Satb2 by competing with Satb2 for the binding to Hdac1, a histone deacetylase normally recruited by the Satb2-NuRD complex bound to the Ctip2 locus (*Alcamo et al., 2008*; *Britanova et al., 2008*). Notably, the distribution of both Lmo4+ and C/S+ cell populations in lower layers change over time in the cortical motor and somatosensory areas during postnatal stages of development.

Overall, we show that common molecular pathways may regulate different specification programs during postnatal maturation and/or refinement of functionally distinct classes of neocortical neurons (such as IT and PT neurons) in an area- and time-specific manner.

## Results

### A consistent subpopulation of cortical lower layer neurons postnatally co-expresses Ctip2 and Satb2

Previous studies reported transient embryonic Ctip2/Satb2 (C/S+) co-expression in the developing cortical plate; however, whether this co-expression persists at postnatal stages was still unclear (*Alcamo et al., 2008*; *Baranek et al., 2012*; *Britanova et al., 2008*; *Leone et al., 2014*). We thus examined the number and distribution of C/S+ cells from E14.5 to P21 to investigate whether they constitute a transient or rather a permanent population of cortical PNs. We first observed that while the number of single Ctip2+ and double C/S+ cells strongly decreases in the cortical plate from E14.5 to E16.5, as previously reported (*Leone et al., 2014*), double C/S+ (but not single Ctip2+) cells resurge between E16.5 and P0 (*Figure 1A and B*) and constitute 18% of lower layer (LL) neurons of the somatosensory cortex (S1) by P21 (*Figure 1B*). Then, to verify whether the postnatal increase of C/S+ neurons was due just to a general increment in Ctip2 and Satb2 expression, we calculated the percentage of C/S+ cells on the total of Ctip2+ or Satb2+ cells and found an increase with respect to both populations (*Figure 1C*).

Next, we investigated whether this population of C/S+ neurons was differentially distributed between frontal/motor (F/M) and parietal/somatosensory cortices. Interestingly, we observed that the highest percentage of C/S+ cells is primordially localized in layer V of the F/M cortex at P0, whereas the corresponding layer of the prospective somatosensory (pS) region has five times less double-positive cells (*Figure 1D*). When calculated on the total of Ctip2+ cells, the percentage of C/S+ neurons only slightly changes between F/M and pS, whereas it results four times higher in the F/M cortex with respect to the total of Satb2+ cells (*Figure 1—figure supplement 1A*). This trend is inverted at P7, where the highest percentage of double C/S+ cells in LL is localized in the primary somatosensory area (S1) (*Figure 1E* and *Figure 1—figure supplement 1B*), suggesting the existence of a time- and region-specific mechanism allowing Ctip2 expression in Satb2+ cells.

Taken together, our data reveal the progressive postnatal and area-specific development of a molecularly hybrid neuronal subpopulation co-expressing two transcriptional regulators with mutually exclusive functions in cortical development.

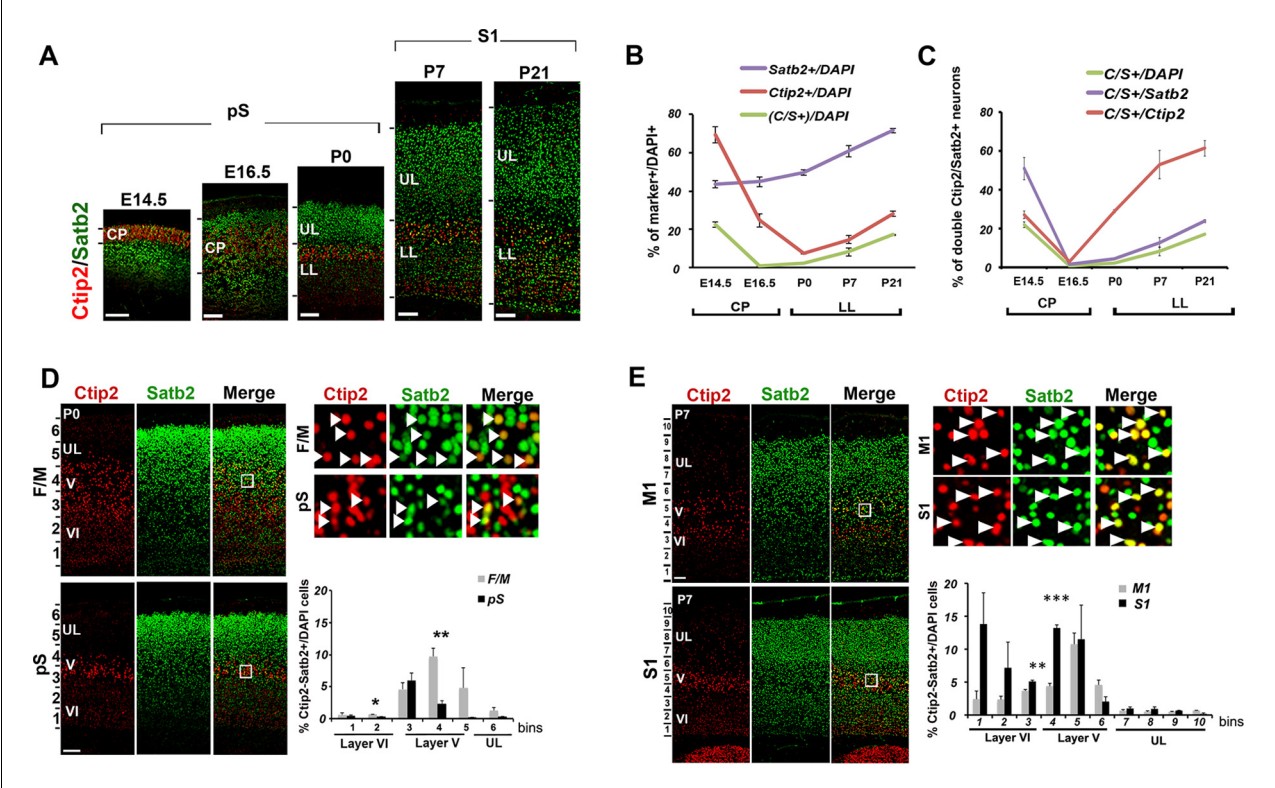

**Figure 1.** Temporal and areal distribution of Ctip2/Satb2+ neurons in the neocortex. (**A**). Coronal sections from prospective (pS) and primary (S1) somatosensory areas of E14.5, E16.5, P0, P7, and P21 cortices immunolabeled for Ctip2 and Satb2. (**B**). Percentage of Satb2+, Ctip2+, and double Ctip2/Satb2 (C/S+) neurons on the total number of DAPI+ neurons in the cerebral cortex at different ages. The counting was performed in the cortical plate (CP) from E14.5 to E16.5 and only in lower layers (LL) from P0 to P21. (**C**). Number of C/S+ neurons calculated as a percentage of DAPI+, Satb2+, and Ctip2+ cortical neurons at different ages. (**D**). Immunostaining for Satb2 and Ctip2 on P0 brain coronal sections from frontal/motor (F/M) and pS areas. Top right panels represent high-magnification views of boxes in layer V depicted in left panels. Arrowheads indicate C/S+ neurons. In bottom right panels, quantification and laminar distribution of double C/S+ neurons. (**E**). Immunostaining for Satb2 and Ctip2 on P7 brain coronal sections from primary motor (M1) and S1 cortices. Top right panels represent high-magnification views of boxes in layer V depicted in left panels. Arrowheads indicate C/S+ neurons. In bottom right panels, quantification and laminar distribution of double C/S+ neurons. Data are represented as means ± SEM. *p≤0.05, **p≤0.01, ***p≤0.001. SEM, standard error of the mean; UL, upper layers.Scale bars: A,D, E, 100 μm.

The following figure supplement is available for figure 1:

**Figure supplement 1.** Time- and area-specific variations in the number and distribution of Ctip2/Satb2+ cells in early postnatal brains.

## Double Ctip2/Satb2+ neurons project to callosal and subcerebral targets

Since double C/S+ neurons are maintained at least until P21, we next investigated the molecular features and connectivity distinguishing these cells from single Ctip2+ or Satb2+ PNs. To analyze their connectivity, we injected cholera toxin subunit B (CTB)-conjugated fluorophores (*Conte et al., 2009*) into the cervical spinal cord, the rostral pontine region and ithe contralateral S1 cortex of P2/P3 pups (*Figure 2A–C*). Injections performed at the level of the spinal cord exclusively labeled cortico-spinal PNs (CSpPNs) (*Figure 2A*), whereas injections in the rostral pons marked all layer V subcerebral PNs (SCPN), including CSpPN, cortico-pontine (CpPN), and other cortico-brainstem (CBPN) neurons projecting through the pyramidal tract (*Figure 2B*). Immunolabeling of retrograde-traced cells at P7 revealed that C/S+ neurons constitute 31% of all subcerebral neurons; however, they do not reach the spinal cord indicating that C/S+ subcerebral neurons most probably project to the brainstem nuclei (*Figure 2A and B*). CTB injections in the contralateral S1 cortex labeled callosal projection neurons (CPN) and showed that 32% of these neurons co-expressed Ctip2 and Satb2 (*Figure 2C*).

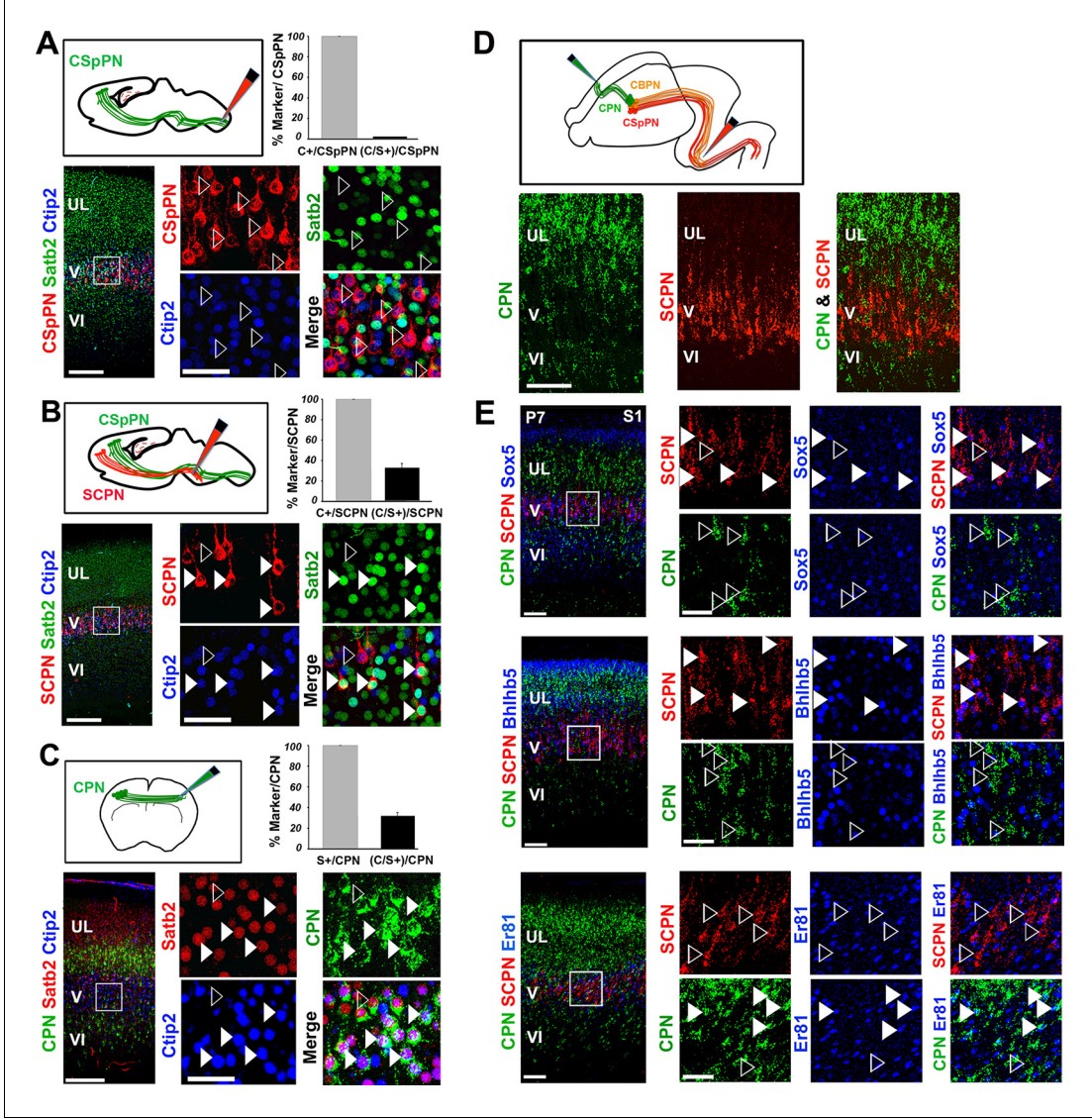

**Figure 2.** Ctip2 and Satb2 co-localization describes subgroups of cortico-brainstem and callosal projection neurons (**A-C**). Top left, schematic representations of cholera toxin subunit B (CTB) injections at cervical level to trace corticospinal projection neurons (CSpPNs) alone (**A**), at the midbrain/hindbrain junction to label subcerebral projection neurons (SCPN) including CSpPNs (**B**), and in the contralateral somatosensory cortex (S1) to label callosal projection neurons (CPN. (**C**). On bottom left of (**A-C**), immunostaining for Satb2 and Ctip2 on retrogradely labeled PNs of P7 S1 coronal sections. Squared panels in (**A-C**) represent high magnification views of boxes in layer V depicted in left panels. Filled arrowheads indicate retrogradely labeled neurons double labeled for Satb2 and Ctip2, while empty arrowheads indicate retrogradely labeled neurons labeled either for Ctip2 or Satb2. On top right of (**A-C**), quantification of Ctip2+ and double Ctip2/Satb2+ (C/S+) retrogradely labeled neurons on the total number of retrogradely labeled PNs in layer V. (**D**). On the top, schematic representation of the labeling paradigm to simultaneously double-label SCPN and CPN. Red retrobeads or Green IX retrobeads were injected into the midbrain/hindbrain junction at P2 and into the contralateral S1 at P3, respectively. On the bottom, panels representing labeled CPN (green) and SCPN (red) on P7 S1 coronal sections. (**E**). Left panel columns represent coronal sections traced for CPN and SCPN and immunostained for Sox5, Bhlhb5, and Er81. Panels on the right represent high-magnification views of boxes in layer V depicted in left panels. Filled white and yellow arrowheads indicate CPN or SCPN positive for the used marker, whereas empty arrowheads indicate negative staining. Scale bars: A,B,C, low-magnification images, 200 μm; D, E, low-magnification images, 100 μm; A,B,C,E, high magnifications, 50 μm. Data are represented as mean ± SEM. SEM, standard error of the mean.

The following figure supplement is available for figure 2:

**Figure supplement 1.** Analysis of the molecular code of postnatal Ctip2/Satb2+ neurons.

The observation that C/S+ neurons project to both contralateral and subcerebral targets raised the question of whether these cells constitute a hybrid population of dual-projecting neurons or two distinct subpopulations. To address this issue, CTB-coated beads were co-injected into the pons and the contralateral S1 of early postnatal brains (*Figure 2D*). No co-labeled callosal and subcerebral cells were observed in three independent experiments (*Figure 2D*), indicating that neurons co-expressing Ctip2 and Satb2 constitute two independent subpopulations: one projecting to the brainstem and the other to the contralateral hemisphere through the corpus callosum.

We then tested whether double C/S+ cells could be further distinguished by the expression of other molecular markers. We used the transcription factors Bhlhb5, highly expressed in CSpPNs of the sensorimotor cortex (*Joshi et al., 2008*), Sox5 present in the major classes of corticofugal PNs (subplate, corticothalamic and all subcerebral PNs) (*Lai et al., 2008*), and Er81(also named Etv1) expressed throughout layer V in both CPNs and SCPNs (*Molnar and Cheung, 2006*; *Yoneshima et al., 2006*). Analyzing the distribution of these factors in C/S+ cells of P7 S1 cortices indicated that nearly 90% of C/S+ cells also express Bhlhb5, 58% Er81 and almost 70% Sox5 in layer V (*Figure 2—figure supplement 1A and B*). To further assess whether they are selective for callosal and/or subcerebral C/S+ cell subpopulations, we combined retrograde labeling with immunostaining for Bhlhb5, Er81, or Sox5 and found that Sox5 and Bhlhb5 are exclusively expressed in SCPNs, whereas Er81 solely labels CPNs in layer V (*Figure 2E*). Taken together, these data indicate that Ctip2/Satb2 co-expression describes at least two layer V subpopulations: callosal PNs co-expressing Er81, and cortico-brainstem PNs co-expressing Bhlhb5 and/or Sox5.

## Morphological and electrophysiological characterization support distinct subpopulations of double Ctip2/Satb2+ neurons in S1 cortex

To unravel specific morphological and electrophysiological features of C/S+ cells, we exploited the *Thy1-eYFP-H* transgenic line, which labels layer Vb neurons from P14 onwards (*Feng et al., 2000*; *Porrero et al., 2010*). We first verified that YFP+ neurons did express Ctip2 and Satb2 in the S1 area of P21 *Thy1-eYFP-H* cortices (*Figure 3—figure supplement 1A*). While 19.3% of GFP+ neurons resulted positive for Ctip2 but not for Satb2 (+/-) and 76.5% co-expressed Ctip2 and Satb2 (+/+), only 1.4% was positive for Satb2 alone (-/+) and 2.8% were negative for both markers (*Figure 3—figure supplement 1A and B*). In addition, YFP+/(C/S+) cells in layer V represent 55.7% of double C/S+ neurons, indicating overall that this mouse line represents an appropriate tool to undertake a detailed morphological and electrophysiological analysis of C/S+ neurons.

Comparison of different morphological features including soma shape, dendritic complexity, and apical dendrite length of YFP+ 3D-reconstructed neurons allowed the classification of C/S+ and single Ctip2+ neurons into two major subpopulations (*Figure 3A and B*). Overall, the soma of C/S+ neurons is significantly smaller in terms of diameter, area, and volume when compared to single Ctip2 neurons; moreover, it occupies on average deeper regions of layer Vand shows earlier bifurcation of the apical tuft. However, K-means clustering of all these parameters revealed that the C/S+ cells are constituted by at least three different subtypes, whereas Ctip2+ neurons by at least two (*Figure 3B–D*). Whereas subtype 1 (orange) is unique to C/S+ neurons, subtype 2 (magenta) and subtype 3 (green) are common to both groups, even if subtype 2 is prevalent in Ctip2+ cells and subtype 3 is mainly represented in C/S+ neurons. Thus, C/S+ and Ctip2+ neurons can be mainly subdivided into two distinct morphological subgroups in the P21 S1 cortex.

Since neurons with a large soma constituted broad fractions of both C/S+ and Ctip2+ populations (*Figure 3—figure supplement 1B*), we investigated whether they would differ for their respective electrophysiological characteristics. We recorded the activity of large soma YFP+ cells on P21 *Thy1-eYFP-H* brain slices and labeled recorded neurons with biocytin to subsequently test them for the expression of Ctip2 and/or Satb2 (*Figure 3E*, *Figure 3—figure supplement 2A*). Steps of hyperpolarizing currents were first applied and the input resistance was measured, based on the I-V curves (*Figure 3F, G and G'*). Cells only expressing Ctip2 (n = 9) had a greater resistance compared to C/S + neurons (n = 23) (Ctip2+: $R_{peak}$ = 109.8 ± 11.7 M$\Omega$ and $R_{ss}$ = 84.8 ± 7.9 M$\Omega$; C/S+: $R_{peak}$ = 73.3 ± 3,9 M$\Omega$ and $R_{ss}$ = 57.1 ± 2.3 M$\Omega$; respectively; p<0.05) as well as a greater sag (difference between the voltage at peak and at steady-state: 4.6 ± 1.2 mV and 2.1 ± 0.3 mV, respectively, p<0.05) (*Figure 3F*), indicating that these two populations with large soma can be discriminated by their intrinsic electrical properties.

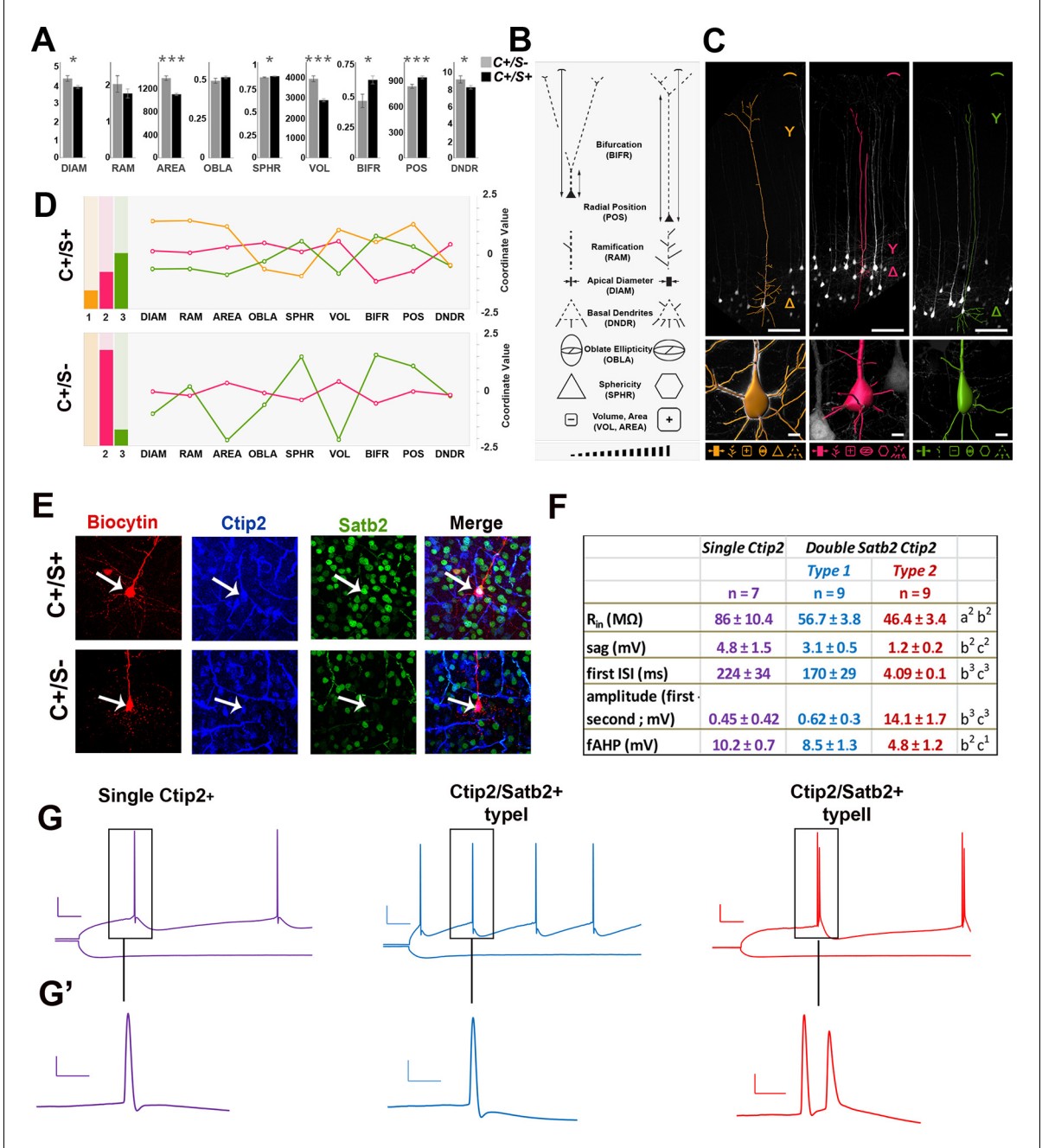

**Figure 3.** Morphometric and electrophysiological characterization of *Thy1-eYFP-H* layer V neurons. (A). The bar charts represent comparisons between double C/S+ (*black*) and single Ctip2+ (*grey*) neurons for different morphological features in YFP+ cells of P21 *Thy1-e-YFP-H* transgenic brains. The asterisks indicate statistical significance. *p≤0.05, **p≤0.01, ***p≤0.001. (B). Pictograms used to schematically illustrate morphological features and qualitative differences among YFP+ neurons. (C). 3D reconstructions of representative neurons of the three distinct morphological profiles. Upper part of the image illustrates length (expressed as soma distance from the pial surface) and bifurcation of the apical dendrite. The bottom part of the images indicates soma reconstruction and basal dendrite data. (D). The bar charts on the left represent the relative number of cells belonging to the morphological profiles identified by K-mean clustering analysis performed separately on each of the two molecular classes (double C/S+ and single Ctip2+ cells). The line graphs on the right represent morphological features for profile 1 (*orange*), unique to double C/S+ cells, profile 2 (*magenta*) and 3 (*green*), shared by both groups. (E). Immunofluorescence for Satb2, Ctip2, and biocytin on S1 coronal sections from P21 *Thy1-eYFP-H* transgenic cortices. (F). Table showing the input resistance (R$_{in}$ reflecting the membrane resistance), the sag (difference of voltage between peak and steady-state potentials), the first interspike interval (ISI) and the difference of amplitude between the first and second action potential (AP), and the fast after-hyperpolarization (fAHP) of the three identified subpopulations. (G). Traces showing the variation in membrane potential when a hyperpolarizing current was injected (-0.2 nA; bottom) and the trains of action potentials when a depolarizing current was injected to the cell to reach the AP threshold (top).

*Figure 3 continued on next page*

*Figure 3 continued*

(**G'**). Magnifications of first and/or second APs. Scale bars: **G**, 20 mV - 50 ms; **G'**, 20 mV - 5ms. Statistics (Mann-Whitney): a = difference between Ctip2+ and C/S+ type 1 cells; b = difference between Ctip2+ and C/S+ type 2; c = difference between C/S+ type 1 and type 2 cells. Data are represented as means ± SEM. [1]p<0.05; [2]p<0.01; [3]p<0.001. SEM, standard error of the mean. Scale bars: **C**, 10x mag., 100 µm; **E**, 40x mag., 10 µm.

The following figure supplements are available for figure 3:

**Figure supplement 1.** Morphometric properties of YFP-positive neurons from the *Thy1-eYFP-H* transgenic line.

**Figure supplement 2.** Electrophysiological analyses of YFP-positive neurons from the *Thy1-eYFP-H* transgenic line.

Interestingly, the analysis of trains of action potentials generated by a step of depolarizing current distinguished again two distinct subpopulations within the C/S+ group. The first type of C/S+ neurons produces a train of single action potentials similar to those obtained in Ctip2+ cells, while the second type generates doublets or even triplets of action potentials (*Figure 3G,G'* and *Figure 3— figure supplement 2B*). Analysis from the I-V curves or from the action potentials generated at threshold showed further differences between the two types of C/S+ neurons, such as the cell resistance and size of the sag or the characteristics of action potentials and inter-spike intervals (*Figure 3F* and *Figure 3—figure supplement 2C*). Taken together, these data support the existence of two subtypes of C/S+ neurons that differ from Ctip2+ cells, and confirm that at P21 neurons co-expressing Ctip2 and Satb2 represent distinct subclasses of cortical PNs in layer V of the S1 cortex.

## Dynamic expression of Lmo4 correlates with Ctip2/Satb2+ cell number and area-specific distribution

We next aimed at deciphering the mechanisms responsible for the co-expression of Ctip2 and Satb2 in the postnatal somatosensory cortex. Satb2 is known to repress Ctip2 expression by recruiting the Nucleosome Remodeling and Deacetylase (NuRD) complex, which in turn deacetylates the Ctip2 locus by interacting with the histone deacetylase 1 (Hdac1) (*Alcamo et al., 2008*; *Britanova et al., 2008*). The protooncogene Ski was shown to play a key role in the Hdac1-NuRD complex interaction (*Baranek et al., 2012*); hence, we hypothesized that the resurgence of C/S+ cells at postnatal stages might be due to the down-regulation of Ski. On the contrary, Ski expression remained high from P0 to P21 and was observed also in several C/S+ cells after birth (*Figure 4—figure supplement 1A–C*), suggesting that a Ski-dependent mechanism is unlikely to contribute to the postnatal increase of C/S+ neurons in the S1 cortex.

Among other candidate genes that might interfere with the Satb2-mediated Ctip2 repression, we selected the transcriptional adaptor Lmo4. This factor interacts with several components of the NuRD complex, is highly expressed in the rostral F/M region, where C/S+ are more abundant, and only in scattered cells of the pS cortex at P0 (*Figure 4A*) (*Cederquist et al., 2013*; *Gomez-Smith et al., 2010*; *Huang et al., 2009*; *Singh et al., 2005*). Lmo4 expression gradually increases in S1 at postnatal stages, reaching its peak in LL at P7, then in all layers at P21 (*Figure 4B*). The timing of Lmo4 expression is thus consistent with the increase of C/S+ cells in S1 from P0 to P21 (*Figure 1B-C* and *4B*), and accordingly, the number of triple Lmo4/Ctip2/Satb2-expressing cells progressively increases from P0 to P21 in layers V and VI (*Figure 4C*). Thus, the temporal and spatial dynamics of Lmo4 expression in C/S+ neurons indicate that this factor might favor their specification at peri- and postnatal stages of corticogenesis.

## Lmo4 affects number and distribution of Ctip2/Satb2+ cells in S1 cortex

To determine whether Lmo4 is required in the specification of C/S+ cells, we first exploited a mouse mutant line (*Lmo4 CKO*), in which *Lmo4* is specifically inactivated in the cortex under the control of the *Emx1* promoter (*Huang et al., 2009*) (*Figure 4—figure supplement 2A*). We examined Ctip2 and Satb2 expression in P7 *Lmo4 CKO* cortices, when normally a high number of S1 lower layer cells and 92% of double C/S+ cells express Lmo4 (*Figure 4B and C*). Notably, the total number of both C/S+ and Ctip2+ cells is decreased in *Lmo4 CKO* S1 cortices (*Figure 4D and E*), whereas the number of Satb2-expressing cells is not particularly affected by the absence of Lmo4 (*Figure 4—figure*

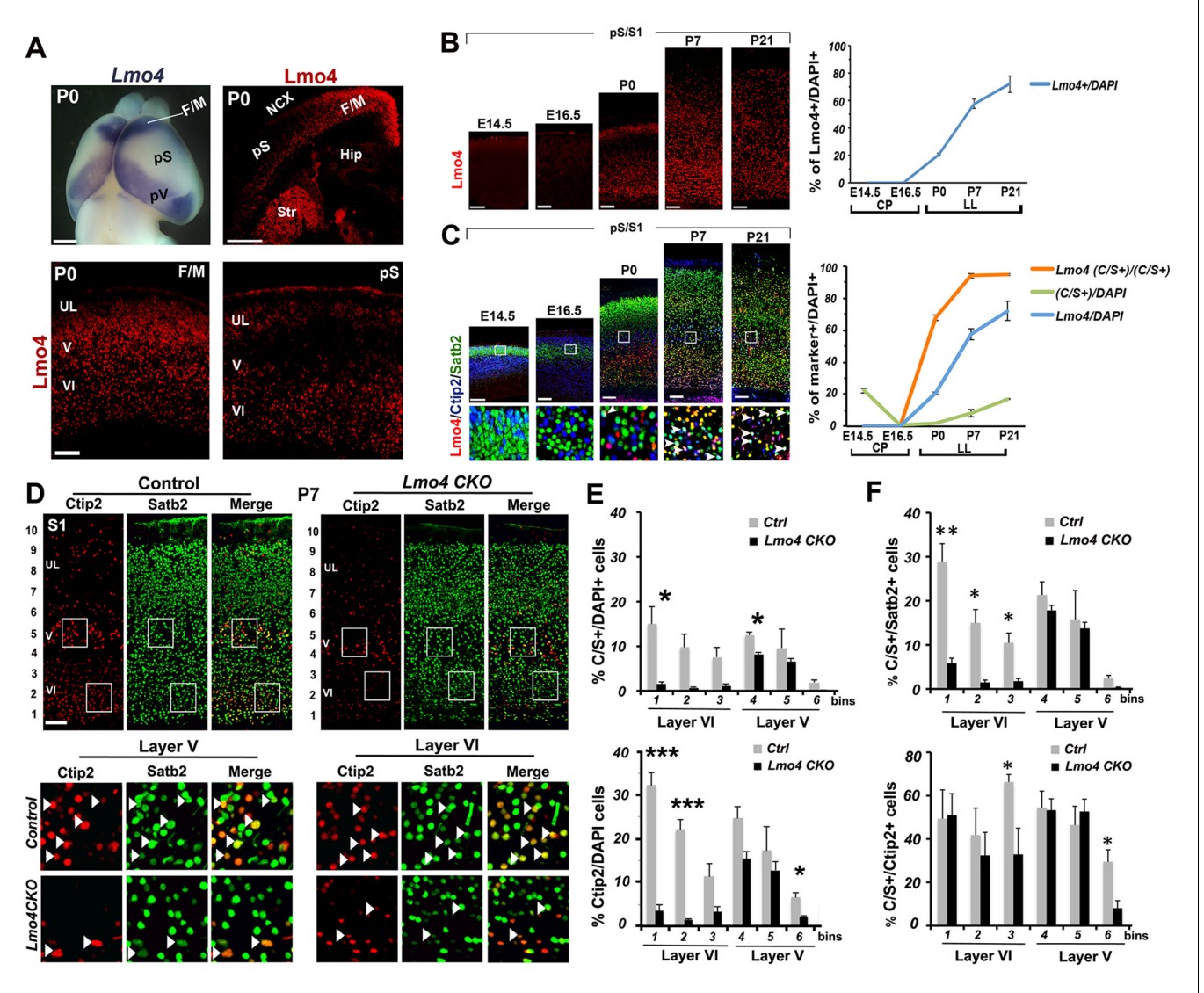

**Figure 4.** Lmo4 controls the number of Ctip2+ and double C/S+ neurons in the somatosensory cortex. (**A**). Whole-mount *in situ* hybridization for *Lmo4* on P0 brain (top left panel) and immunofluorescence for Lmo4 on coronal sections. (Below) Expression of Lmo4 in high-magnification views of frontal motor (F/M) and prospective somatosensory (pS) coronal sections. (**B**). Coronal sections from pS and primary somatosensory area (S1) of brains from E14.5 to P21 immunostained for Lmo4. On the right, time course of the percentage of Lmo4+ neurons on the total of DAPI+ cells (cortical plate -CP- for prenatal brains and lower layers –LL- for postnatal brains). (**C**). Triple immunostaining for Lmo4/Ctip2/Satb2 on coronal sections from pS and S1 E14.5 to P21 brains. Bottom squared panels represent high-magnification views of boxes in layer V depicted in top panels. On the right, quantification of Lmo4+ (*blue line*) and Ctip2/Satb2+ (C/S+) cells (*green line*) on the total of DAPI+ cortical cells, and of triple Lmo4/C/S+ cells (*orange*) on the total number of C/S+ neurons. (**D**). Double immunofluorescence for Satb2 and Ctip2 on coronal sections of P7 control and *Lmo4 CKO* somatosensory (S1) areas. Bottom squared panels represent high-magnification views of boxes in layers V and VI of top panels. Arrowheads point to double C/S+ neurons. (**E** and **F**) Quantification and layer distribution of double C/S+ or single Ctip2+ neurons on the total of DAPI+ cells (**E**), and of double C/S+ on the total of Satb2+ (top panel **F**) or Ctip2+ (bottom panel **F**) neurons. NCX: neocortex, pV: prospective visual, Hip: hippocampus, Str: striatum, UL, upper layer neurons. Scale bars: B, C, D and lower panels of A, 100 μm, upper panels of A, 1mm. Data are represented as means ± SEM. *p≤0.05, **p≤0.01, ***p≤0.001. SEM, standard error of the mean.

The following figure supplements are available for figure 4:

**Figure supplement 1.** Ski is expressed by several C/S+ neurons at postnatal stages.

**Figure supplement 2.** Unaltered distribution of Satb2+ neurons in the absence of Lmo4.

*supplement 2B*). This might indicate that the alteration in C/S+ cells is mainly due to a general decrease in Ctip2 expression. However, the ratio of C/S+ cells decreased also compared to the total number of Ctip2+ (in upper layer V and VI) and Satb2+ cells (in layer VI) (*Figure 4F*). Since the number and distribution of Satb2+ cells does not significantly change, the reduced number of C/S+ cells is most probably due to an increased repression of Ctip2 in Satb2+ neurons.

Next, we investigated whether Lmo4 is cell-autonomously required in the specification of double C/S+ cells in layer V by overexpressing *Lmo4* in the S1 of WT cortices. To this aim, we cloned the coding sequence of *Lmo4* into the *pCdk5r1-IRES-EGFP* vector (*Figure 5A*), which drives the selective expression of a given transcript in postmitotic neurons (*Wang et al., 2007b*). To evaluate the efficacy of this new construct (*pCdk5r1-Lmo4-EGFP*), we electroporated mouse brains at E13.5 and analyzed the somatosensory cortex at P0 when Lmo4 is only faintly expressed (*Figure 4A*). Electroporated (GFP+) cells expressed high levels of Lmo4, whereas similar regions in the contralateral cortex contained just few Lmo4+ cells (*Figure 5B,C*).

The *pCdk5r1-Lmo4-EGFP* construct was then electroporated at E13.5, and brains were collected at P7, when the laminar specification of PNs is nearly completed (*Figure 5D*). Control (*pCdk5r1-EGFP* electroporated) cells were predominantly Satb2+ and Ctip2- in layer V (*Figure 5D*), whereas the overexpression of Lmo4 increased the ratio of Ctip2+ neurons among GFP+ cells from 7.6 ± 4.7% to 31.3 ± 3.8% (n = 3, p = 0.03) and the percentage of C/S+ neurons from 5.2 ± 3.4% to 16.4 ± 1.0% in LL (n = 3, p<0.05) (*Figure 5E*), supporting a cell-autonomous role for Lmo4 in inducing Ctip2 expression in both Satb2+ and Satb2- cells.

Overall, our Lmo4 loss- and gain-of-function approaches both demonstrate that Lmo4 acts in the specification of C/S+ cells primarily by modulating Ctip2 expression in layer V.

## Upregulation of Lmo4 increases the number of double Ctip2/Satb2+ cells in *Nr2f1*-deficient somatosensory cortices

To further confirm that Lmo4 plays a role in the specification of C/S+ cells, we analyzed the distribution of these cells in mice lacking the transcription factor *Nr2f1* (also called *COUP-TFI*) in cortical neurons (*Nr2f1 fl/fl^Emx1-Cre*, from now on *Nr2f1 CKO*) (*Armentano et al., 2007*). *Nr2f1 CKOs* exhibit a remarkable upregulation of Lmo4 in layer V and in the lower part of UL of the somatosensory cortex at P0 (*Alfano et al., 2014*) (*Figure 6A–B*). In agreement with our previous studies (*Alfano et al., 2014*; *Tomassy et al., 2010*), the number of Ctip2+ cells is significantly increased in the mutant motorized somatosensory (mS) cortex, whereas Satb2+ cells are only slightly augmented (*Figure 6C–D* and *Figure 6—figure supplement 1A*). As expected, the number and distribution of C/S+ neurons are also increased on the total of cells and relative to the number of Satb2+ and Ctip2 + cells in *Nr2f1 CKOs* (*Figure 6C–D* and *Figure 6—figure supplement 1A*). Finally, mutant brains showed a much higher number of triple Lmo4/Ctip2/Satb2+ cells in layer V and upper layer VI than controls (*Figure 6—figure supplement 1C*). While 68% of C/S+ cells express Lmo4 in control cortices at P0, this ratio rises to 92% in mutant brains (*Figure 6—figure supplement 1D*), supporting a correlation between increased Lmo4 expression and higher number of C/S+ cells in LL of mutant cortices.

Next, we investigated whether ectopic Lmo4 expression and increased number of C/S+ cells in the mutant somatosensory cortex coincided with altered neuronal connectivity of layer V neurons. Previous work showed that corticospinal projection neurons (CSpPNs) were abnormally located in layer VIa of *Nr2f1 CKO* (*Tomassy et al., 2010*). By injecting CTB-coated beads either into the brainstem or the spinal cord regions of control and *Nr2f1* mutant brains, we observed an expansion of cortico-brainstem projection neurons (CBPNs) in layer Va and confirmed the mispositioning of the CSpPNs in upper layer VI of mutant cortices (*Figure 6E*). Interestingly, the highest percentage (66.4 ± 6.7% ) of CBPNs co-expressing Ctip2 and Satb2 was in layer Va, whereas none of the labeled cells co-expressed these genes in layer VIa (*Figure 6F and G*). This suggests that Lmo4 increase may underlie both the strong increase in C/S+ cells and the shift of connectivity from corticospinal to corticobrainstem targets observed in mutant layer V.

Thus, to demonstrate that the upregulation of Lmo4 in the *Nr2f1 CKO* brains was directly involved in layer V Ctip2 radial expansion, we electroporated an *Lmo4-specific shRNA* construct (*Qin et al., 2012*) into E13.5 mutant brains and analyzed Lmo4 and Ctip2 laminar distribution in P0 electroporated mS1 cortices (*Figure 6—figure supplement 2*). While GFP+ control cells co-localize with Lmo4 and Ctip2 (*Figure 6—figure supplement 2A,B*), *Lmo4 shRNA*-expressing cells show a

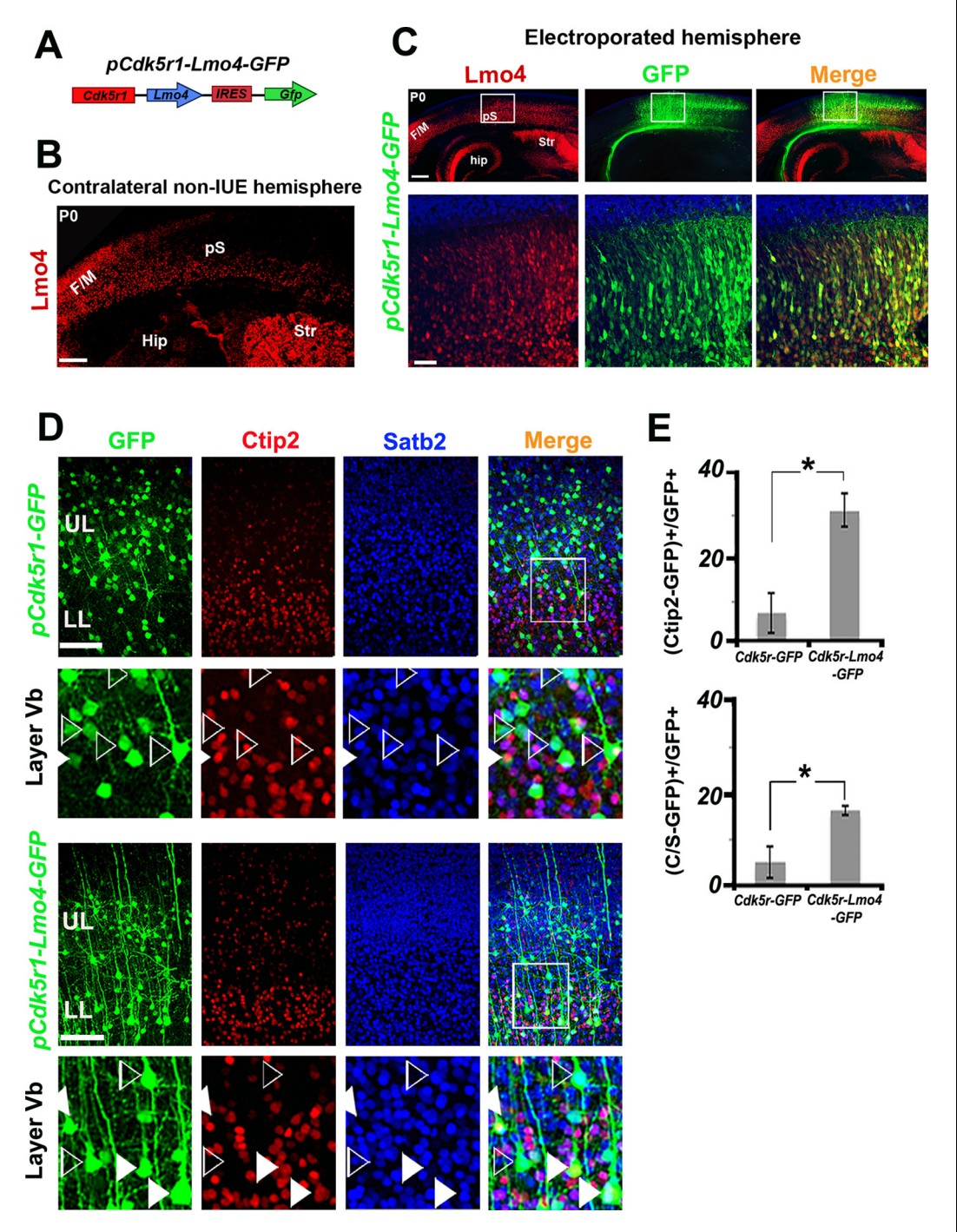

**Figure 5.** Lmo4 overexpression increases the number of Ctip2+ and C/S+ neurons in lower layers. (A). Schematic representation of the vector used to overexpress *Lmo4* in postmitotic neurons. (B). Immunostaining for Lmo4 on a coronal section of the contralateral (non-electroporated) hemisphere of P0 electroporated brains. (C). Coronal sections of a *pCdk5r1-Lmo4-IRES-GFP* electroporated E13.5 hemisphere immunolabeled for Lmo4 and GFP at P0. Bottom squared panels represent high-magnification views of boxes depicted in upper panels. (D). Immunostaining for Satb2, Ctip2, and GFP on coronal sections from P7 S1 cortices electroporated at E13.5 with *pCdk5r1-IRES-GFP* (on the top) or *pCdk5r1-Lmo4-IRES-GFP* (on the bottom). Squared panels represent high-magnification views of boxes depicted in top panels. Filled arrowheads indicate double C/S+ GFP+ cells, whereas empty arrowheads indicate GFP+ cells not expressing Ctip2. (E). Quantification of Ctip2+/GFP+ cells on the total number of GFP+ cells (on the top), and of (C/S+)/GFP+ cells on the total number of GFP+ cells (on the bottom) in layer V of electroporated brains. Data are represented as means ± SEM. *p≤0.05. (IUE) *In utero* electroporated; (Hip) hippocampus; (F/M) frontal motor area; (pS) prospective somatosensory area; SEM, standard error of the mean. Scale bars: **B,C** 300 µm; **C,** high magnification: 50 µm, **D**, 100 µm.

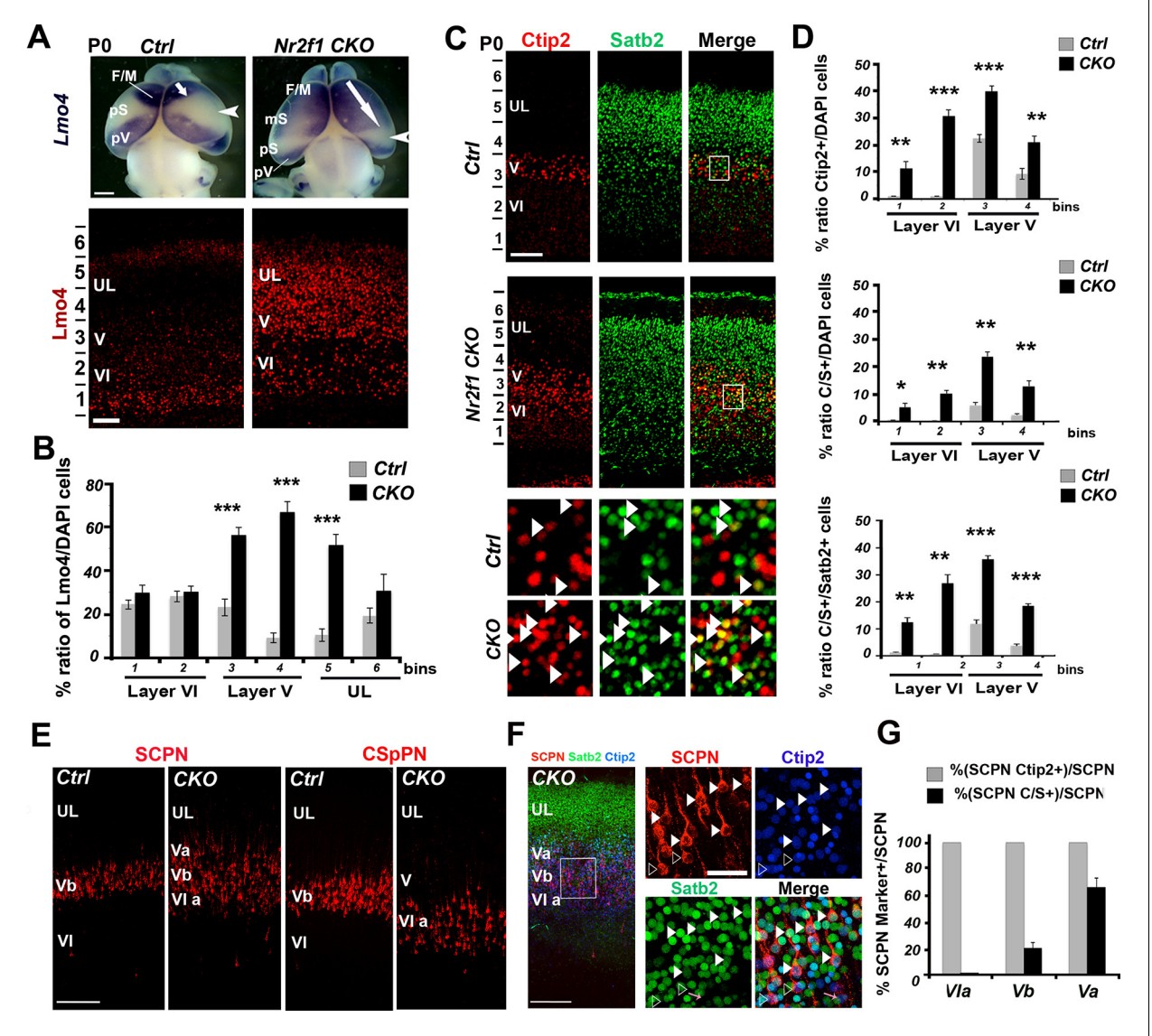

**Figure 6.** Increase of Lmo4- and double C/S-expressing neurons in the motorized somatosensory cortex of Nr2f1 CKO brains. (**A**). Whole-mount *in situ* hybridization for *Lmo4* (top panels) and Lmo4 immunofluorescence on coronal sections from the prospective somatosensory (pS) cortex (bottom panels) of P0 control and *Nr2f1 CKO* brains. (**B**). Quantification and layer distribution of Lmo4+ neurons in pS of P0 control and *Nr2f1 CKO* brains. (**C**). Coronal sections from the pS of P0 control and *Nr2f1 CKO* brains immunolabeled for Ctip2 and Satb2. Bottom squared panels represent high-magnification views of layer V neurons in boxes depicted in top panels. Arrowheads indicate C/S+ neurons. (**D**). Quantification of Ctip2+ and C/S+ neurons in the pS of P0 control and *Nr2f1 CKO* brains as a percentage of the total number of cells (DAPI+) or of Satb2+ neurons. (**E**). Coronal sections of S1 from controls and *Nr2f1 CKO* P7 brains retrogradely-labeled in the pontine region and in the cervical spinal cord. (**F**). Immunofluorescence for Satb2 and Ctip2 on P7 *Nr2f1 CKO* S1 retrogradely-labeled cortices. Filled arrowheads in high-magnification views indicate retrogradely-labeled subcerebral projection neurons (SCPNs) double positive for Satb2 and Ctip2, whereas empty arrowheads indicate retrogradely-labeled SCPNs positive for Ctip2. (**G**). Quantification of Ctip2+ and C/S+ retrogradelylabeled SCPNs on the total number of labeled PNs in layers Va, Vb, and VIa of *Nr2f1 CKO* brains. F/M, frontal motor area; pS, prospective primary somatosensory area; pV, prospective primary visual area; CSpPN, corticospinal projection neurons. UL, upper layers; VI, layer VI. Data are represented as means ± SEM. *p≤0.05, **p≤0.01, ***p≤0.001. SEM, standard error of the mean. Scale bars: A, 1 mm; lower panel A, C, 100 μm; E, F, 200 μm; high-magnification views in F, 50 μm.

The following figure supplements are available for figure 6:

**Figure supplement 1.** Increase of double Ctip2/Satb2+ and triple Lmo4/Ctip2/Satb2+ neurons in the motorized *Nr2f1 CKO* somatosensory cortex.

**Figure supplement 2.** Decrease of Ctip2 expression in the motorized *Nr2f1 CKO* somatosensory cortex after Lmo4 downregulation.

drastic reduction not only in Lmo4 expression, as expected, but also in Ctip2 levels in layer Va (*Figure 6—figure supplement 2C,D*). This suggests that Ctip2 overexpression in layer Va is mainly due to Lmo4 increase and that *Nr2f1* loss not only upregulates Lmo4 levels but also favors its action on Ctip2 expression. In addition, cells electroporated with the control *shRNA* show a delayed migration to the apical region of the cortex, a phenotype that was already described in our previous publication (*Alfano et al., 2011*). Notably, downregulating *Lmo4* rescues the radial migratory defect and allows electroporated cells to reach the proper radial position in the upper regions of P0 cortices.

These data demonstrate that Lmo4 is functionally involved in the abnormal upregulation of layer V Ctip2 expression observed in the absence of Nr2f1, and more in general, support a key role for Lmo4 in Ctip2 de-repression.

## Lmo4 de-represses Ctip2 by competing with Satb2 for Hdac1 binding

Next, we investigated whether Lmo4 was able to positively regulate Ctip2 by directly interfering with the molecular machinery underlying Satb2-mediated Ctip2 repression (*Alcamo et al., 2008*; *Britanova et al., 2008*). Lmo4 was shown to bind Histone deacetylases 1 and 2 (Hdac1 and 2) to repress its downstream target genes (*Singh et al., 2005*). Since the Hdac1-NuRD-Satb2 complex assembly is fundamental for Ctip2 repression (*Alcamo et al., 2008*; *Baranek et al., 2012*; *Britanova et al., 2008*), we examined whether Lmo4 was able to compete with Satb2 for Hdac1 interaction. To this aim, we performed immunoprecipitation with Lmo4- and Satb2-specific antibodies on nuclear proteins extracted from control and *Nr2f1 CKO* P1 cortices, where Lmo4 and Ctip2 are strongly upregulated (*Figure 7A*). Immunoprecipitated protein fractions were analyzed by Western blot using an antibody specific for Hdac1 (*Alcamo et al., 2008*; *Britanova et al., 2008*; *Gyorgy et al., 2008*). Comparable levels of Hdac1 were co-immunoprecipitated (Co-IP) with Lmo4 and Satb2 antibodies in control conditions, whereas Hdac1-Lmo4 interaction increased at the expense of Hdac1-Satb2 in *Nr2f1 CKO* brains. These changes in the amount of each complex were not due to dramatic changes in the total amount of Satb2 and Lmo4 proteins between wt and *Nr2f1* mutant brains. Indeed, Satb2 protein levels resulted unaltered between the two conditions, whereas the amount of Lmo4 resulted only slightly increased in *Nr2f1* mutants (*Figure 7A*). Since the overall Lmo4 upregulation is not remarkable and takes place particularly in the mS1 cortex, the higher amount of Hdac1-Lmo4 complex in mutant Co-IP fractions might be due to a higher binding affinity between these factors in somatosensory regions. Accordingly, the number of C/S+ cells is remarkably higher in lower layers of S1 than in those of M1 in P7 wt brains (*Figure 1E*) despite similar Lmo4 levels between S1 and M1 areas (our data and (*Huang et al., 2009*).

Finally, we investigated whether Lmo4 binds Hdac1 by interacting with the Ski-Hdac1 complex and preventing its interaction with Satb2 (*Baranek et al., 2012*). To this aim, we analyzed the Lmo4-immunoprecipitated nuclear fractions from control and mutant cortices with the antibodies for Ski and Satb2. We found that Lmo4 binds Ski consistently, whereas it interacts very limitedly with Satb2, suggesting that Lmo4 normally interacts with Ski before it binds to the Satb2-NuRD complex. However, the increased Hdac1-Lmo4 binding does not seem to be related to higher interaction between Lmo4 and Ski, since the amount of bound Ski does not change in the mutant extracts (*Figure 7B*).

We next verified whether the chromatin state in the *Ctip2* genetic locus varies between control and mutant cortices (*Figure 7C*). We carried out a chromatin immunoprecipitation (ChIP) assay with the anti-Hdac1 antibody and with an antibody specific for the acetylated form of histone 4 (H4K12ac), whose levels are proportional to the rate of transcriptional activity. Immunoprecipitated chromatin fractions from P1 cortices were analyzed by semi-quantitative PCR and confirmed by QPCR using primers amplifying part of the matrix attachment region (MAR) in the *Ctip2* locus (*Britanova et al., 2008*). As a control, we used primers amplifying a sequence in the last intron of the *Rnd2* gene containing a previously described Nr2f1-binding site (*Alfano et al., 2011*). Our data show that a lower amount of Hdac1 binds the *Ctip2* locus, which accordingly appears more acetylated and thereby more active (*Figure 7C*). No Hdac1 binding and acetylation differences were observed with control primers, confirming specificity of its effect on the *Ctip2* locus (*Figure 7C*).

Since changes in the Hdac1-Satb2 interaction and in Ctip2 de-repression observed in *Nr2f1* mutant brains might be not only or not directly related to the observed Lmo4 increase in the mS1 region, we repeated the CoIP experiments by overexpressing Lmo4 and Satb2 in COS7 cells. To confirm the competition between Lmo4 and Satb2 for Hdac1 binding, COS7 cells were transfected with increasing quantities of a vector expressing *Lmo4* under the control a CMV enhancer (*pCIG2-*

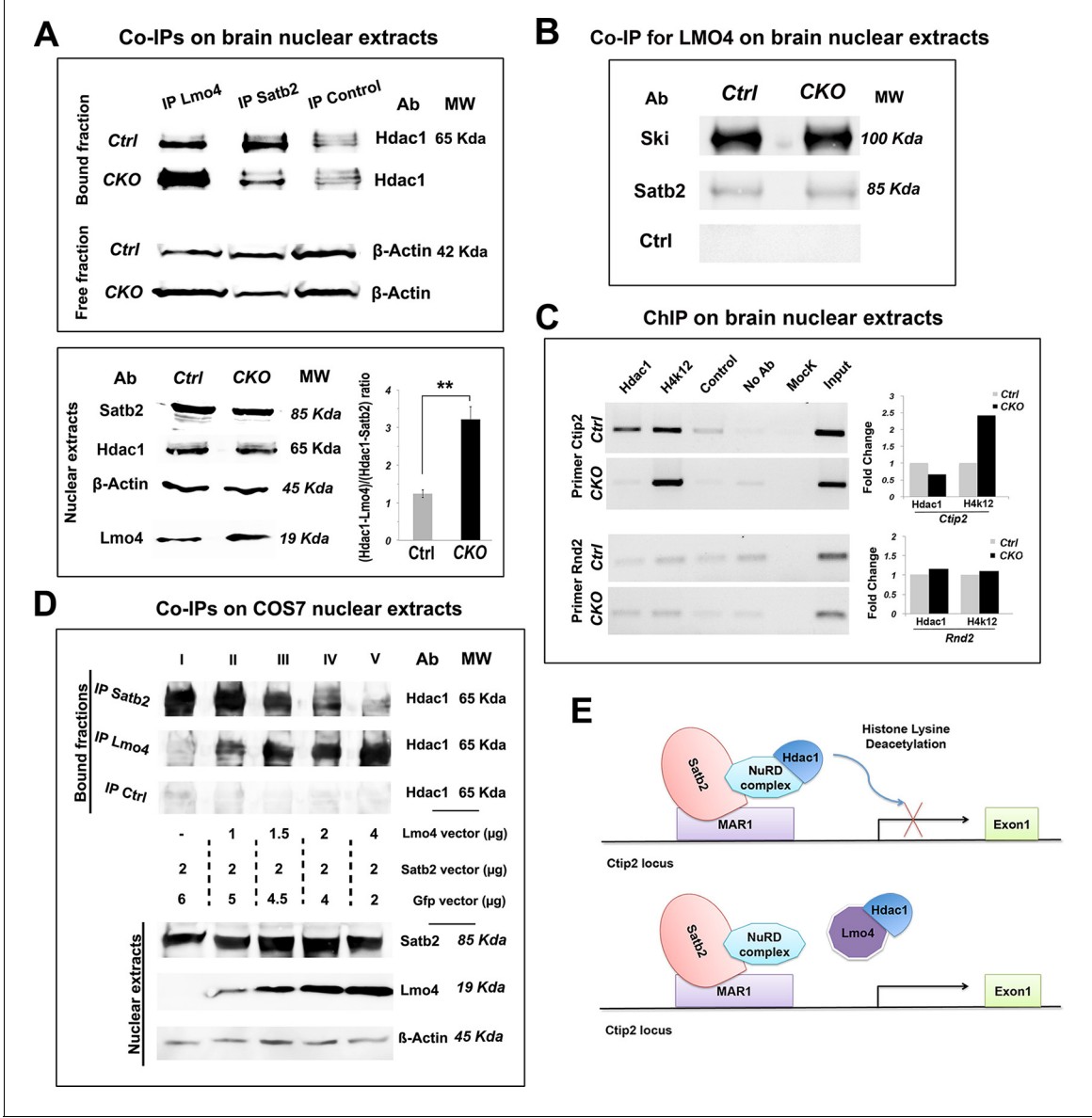

**Figure 7.** Lmo4 interacts with Hdac1 and prevents its binding to the Ctip2 locus. (A). On the top, Western blot on nuclear extracts from controls and *Nr2f1 CKO* P1 cortices immunoprecipitated with antibodies specific for Lmo4 (IP Lmo4), Satb2 (IP Satb2), or an unrelated epitope (IP control). IP fractions were analyzed using an antibody against Hdac1, whereas the corresponding input fractions were analyzed with an antibody specific for β-actin. On the bottom-left, Western blot on nuclear extracts from controls and *Nr2f1 CKO* P1 cortices for Satb2, Hdac1, Lmo4, and β-actin. To the right, ratio between Hdac1-Lmo4 and Hdac1-Satb2 complexes immunoprecipitated from control and *Nr2f1 CKO* extracts. (B). Western blot performed on nuclear extracts from controls and *Nr2f1 CKO* P1 cortices immunoprecipitated with specific antibody for Lmo4. IP fractions were analyzed using an antibody against Ski, Satb2 or an unrelated epitope (control). (C). On the left, semi-quantitative PCR performed on Chromatin-immunoprecipitation (ChIP) samples from controls and *Nr2f1 CKO* P1 cortices. The assay was performed using antibodies against Hdac1, the anti-Histone H4 (acetyl K12) (H4K12) and primers amplifying a MAR1 sequence on *Ctip2* and *Rnd2* loci. On the right, QPCR performed on ChIP samples from controls and *Nr2f1 CKO* cortices. (D). On the top, Western blot on nuclear extracts from COS7 cells transfected with an equal amount of Satb2- and an increasing amount of Lmo4-expressing vectors and immunoprecipitated with antibodies specific for Satb2 (IP Satb2), Lmo4 (IP Lmo4), or an unrelated epitope (IP control). IP fractions were analyzed using an antibody against Hdac1. On the bottom, Western blot on nuclear extracts from the transfected COS7 cells with specific antibodies for Satb2, Lmo4, and β-actin showing increase of Lmo4 and similar Satb2 and β-actin protein levels. (E). Schematic model of the putative mechanism by which Lmo4 de-represses Ctip2 expression. . **p≤0.01. MAR1: Matrix attachment region 1; NuRD complex: Nucleosome Remodeling and Deacetylase complex.

*Lmo4*) together with a constant amount of the *pCAG-Satb2* plasmid (*Britanova et al., 2008*) (*Figure 7D*). Cell extracts, analyzed by immunoprecipitating proteins with an anti-Hdac1 antibody, showed a progressive increase in Hdac1-Lmo4 interaction, in line with the Lmo4 increase, at the expense of Hdac1-Satb2 binding (*Figure 7D*). This confirmed the competition model and put in direct relation the Lmo4 increase with the decreased Hdac1-Satb2 interaction.

Overall, our analysis indicates that Lmo4 progressively interferes with the Satb2-mediated Ctip2 repression by sequestering Hdac1 before it interacts with the Satb2-NuRD complex on the *Ctip2* locus and hence, favoring Ctip2 and Satb2 co-localization in cortical LL during postnatal stages of development (*Figure 7E*).

## Discussion

This study shows that key developmental regulators with opposite functions during embryonic stages, such as Ctip2 and Satb2, can co-localize at postnatal stages and participate in generating the great diversity of PN subtypes in the young adult brain. The transcriptional adaptor Lmo4 epigenetically modifies the *Ctip2* locus and enables *Ctip2* expression in Satb2+ lower layer neurons by interfering with the Satb2-mediated deacetylation in a time- and area-specific manner. Double C/S+ neurons comprise at least two distinct neuronal subclasses with unique connectivity, morphology and electrophysiological profiles in the juvenile (P21) mouse brain.

### Ctip2 and Satb2 co-expression correlates with several neuronal features

After birth, Satb2 and Ctip2 are not any more an exclusive hallmark of callosal (CPN) or subcerebral (SCPN) projection neurons, respectively, but are co-expressed in distinct subclasses of CPNs and SCPNs with specific connectivity profiles, morphological, and electrophysiological characteristics. In line with our results, Satb2 expression was shown to be associated with the SCPN markers Fezf2 and Sox5 in CPN populations of motor areas (*Sohur et al., 2014*; *Tantirigama et al., 2014*). In addition, Satb2 is not only required for CPNs but also for the proper differentiation and axon pathfinding of SCPNs (*Leone et al., 2014*). It is thus possible that early-born CPNs, which originate at a similar time as SCPNs and reach lower layers in S1, are more similar to subcerebral than late-born CPNs of upper layers.

We also show that C/S+ neurons expressing Er81 define a distinct subpopulation of CPNs residing in layer V of S1. This subpopulation is unlikely to project to the contralateral striatum, since callosal-striatal neurons are almost absent in S1 after P15 and fail to express Ctip2 but are instead positive for Sox5 (*Sohur et al., 2014*). Accordingly, none of the C/S+ CPNs identified here were positive for Sox5, whereas a group of C/S+ SCPNs clearly expressed Sox5. Our molecular analysis thus identified novel subclasses of SCPNs and CPNs major neuronal groups with distinct features, which can be specifically identified by Ctip2 andSatb2 co-expression.

The morphological and electrophysiological characterizations revealed hybrid but also unique features of C/S+ subpopulations compared to single Ctip2+ cells. Although a subset of both populations projects to subcerebral targets, the C/S+ neuronal axons do not reach the spinal cord and show typical morphological features of CPNs, such as a small soma, long, and thin apical dendrites and late bifurcation of the apical tuft (*Hattox and Nelson, 2007*). However, a small group of C/S+ neurons also share characteristics of SCPNs with single Ctip2+ cells, such as a large soma, high number of secondary dendrites, thick apical dendrites and an early bifurcation of the apical tuft (*De la Rossa et al., 2013*; *Hattox and Nelson, 2007*; *Tantirigama et al., 2014*). Thus, C/S+ cells include different subsets of PNs sharing some typical CPN and SCPN morphological features.

Electrophysiological recordings also revealed that C/S+ neurons exhibit a lower sag value than single Ctip2+ cells, indicating that C/S+ cells might acquire some UL electrical features, since normally these neurons have a lower hyperpolarization-activated current (Ih) than LL neurons (*De la Rossa et al., 2013*; *Sheets et al., 2011*). This is also reminiscent of the observations made in *Satb2* mutant or *Fezf2*-overexpressing brains, where UL neurons acquire electrical features of LL neurons (*De la Rossa et al., 2013*; *Leone et al., 2014*). Moreover, Ctip2+ neurons show higher ISI (first interspike interval) value than C/S+ cells. A similar variation in ISI values was previously described between UL and LL neurons (*De la Rossa et al., 2013*), suggesting that the co-expression of Satb2 in Ctip2+ layer V neurons may result in electrical features characteristic of UL neurons. Finally, since

these variations in electrophysiological properties were observed by analyzing cells with similar morphologies, C/S+ and Ctip2+ neurons might diverge in the expression of ion channels and pumps (*Oswald et al., 2013*; *Staff et al., 2000*). In the future, it might be interesting to investigate eventual correlations between the expression of Satb2 and/or Lmo4 and the function/expression of specific ion channels in layer V neurons.

## Lmo4 interferes with Satb2-mediated Ctip2 repression in the postnatal cortex

Our time-course analysis of C/S+ neurons confirms that only a small population of cells maintains the expression of these two antithetic factors during perinatal stages of corticogenesis (*Baranek et al., 2012*; *Leone et al., 2014*) (*Figure 8*). This is consistent with the paradigm that competing molecular programs direct the differentiation of major PN classes during late embryonic stages of corticogenesis (reviewed in *Greig et al., 2013*). However, we found that distinct subpopulations of C/S+ PNs are maintained and gradually increase postnatally in lower layers of the S1 cortex.

This study also demonstrates that Lmo4 progressively increases and co-localizes with double C/S + neurons after birth in the somatosensory area. Lmo4 is a widely expressed transcriptional modulator known to regulate several key biological processes, from cell growth to fate determination (*Sang et al., 2014*). It is well known that Lmo4 acts as a scaffolding protein for the assembly of multi-protein complexes and interacts with several co-factors of the NuRD complex (*Gomez-Smith et al., 2010*; *Singh et al., 2005*; *Wang et al., 2007a*).

Although other factors might be involved in Satb2/Ctip2 co-expression, our work demonstrates that Lmo4 de-represses Ctip2 by sequestering Hdac1, a critical component of the NuRD complex recruited by Satb2 on the *Ctip2* locus to inactivate its transcription. Overall, our data constitute first direct evidence that the control of epigenetic mechanisms may underlie area-specific variations in neuronal features. Most importantly, such processes take place after birth and seem to contribute to the maturation and refinement rather than to the initial specification of neuronal subtypes of the somatosensory area.

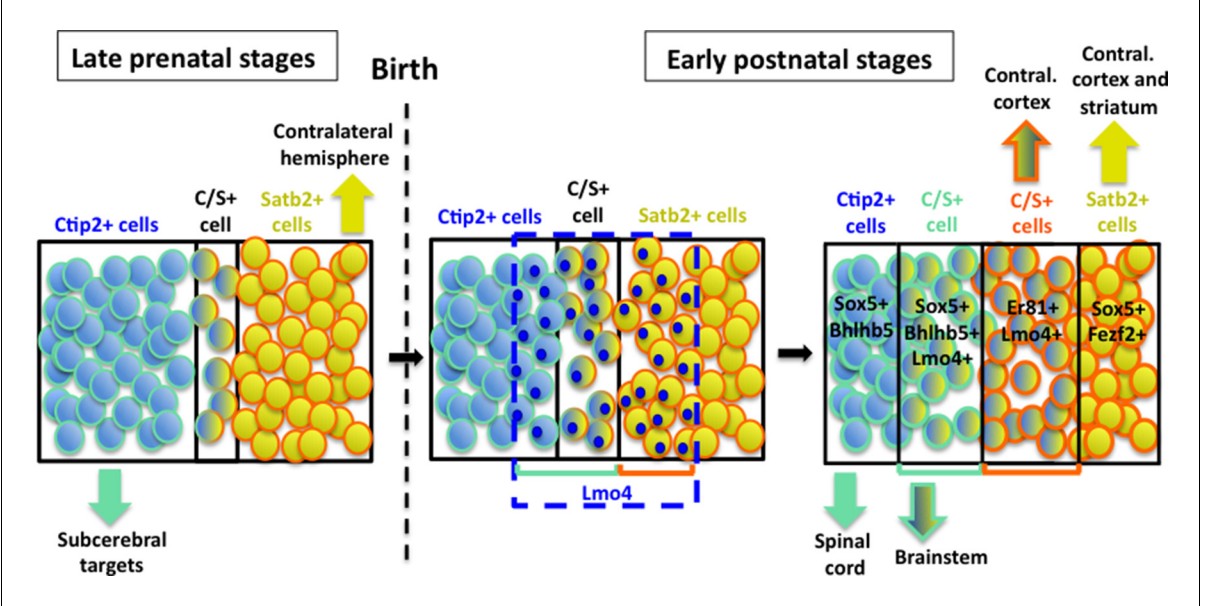

**Figure 8.** Hypothetical model of layer V neuronal subtype refinement. After birth, Lmo4 expression progressively increases in Satb2+ and in Ctip2+ cells of S1 cortex. Lmo4 de-represses Ctip2 expression in Satb2+ cells and probably Satb2 in Ctip2+ cells. According to the different dynamics leading to the refinement of distinct C/S+ subclasses, these cells will project either to subcerebral or to contralateral targets. Vertically oriented arrows indicate connectivity at postnatal stages. The blue dashed line and blue circles in different cells delimit territories of Lmo4 expression. Blue marine circles surrounding cells depict subcerebral projection neuron (SCPN) subpopulations, whereas orange ones delineate callosal neuron (CPN) subgroups. Black lines delimit different neuronal populations of the somatosensory cortex.

Obviously, this study does not constitute an exhaustive investigation on the molecular mechanisms regulated by Lmo4, which seems to be also able to induce Satb2 expression in Ctip2-positive cells, since a notable portion of C/S+ cells seems to derive from a subpopulation of Ctip2+ cells (model in *Figure 8*). This is not totally unexpected in light of previous reports that correlated Lmo4 expression with callosal (Satb2+) development (*Azim et al., 2009*; *Molnar and Cheung, 2006*; *Ye et al., 2015*). Accordingly, we found that the number of C/S+ cells decreased in *Lmo4* mutants not only on the total of the Satb2+ subpopulation but also on the total of the Ctip2+ one, suggesting that C/S+ cells may also derive from Lmo4-mediated de-repression (and/or activation) of Satb2 in Ctip2+ cells.

Finally, we noticed that Ctip2 was poorly upregulated in upper layers of *Nr2f1 CKO* motorized somatosensory cortices and of *Lmo4*-electroporated cortices. This could be explained by the presence of a laminar-specific transcription factor, which would act synergistically with Lmo4 to efficiently activate Ctip2 expression in lower layers. More experiments are required to test this hypothesis.

### Possible mechanisms of action and interactors of Lmo4, Ctip2, and Satb2 during SCPN maturation in the somatosensory area

Lmo4 had already been implicated in specifying SCPN and CPN identities in the rostral motor cortex (*Cederquist et al., 2013*), but a similar function in sensory areas was not previously unveiled. Here, we show that Lmo4 acts preferentially in layers V and VI of the S1 cortex in which the majority of neurons (SCPN) projects subcerebrally and expresses Ctip2 at different levels. Lmo4 levels gradually increase from layer VI to upper layers at postnatal stages, together with the progressive specification of C/S+ cells.

Interestingly, we report that C/S+ neuron projections target the brainstem, but not the spinal cord. If Lmo4 and/or Satb2 and Ctip2 co-localization modulates axon targeting, what are their mechanisms of action? Ctip2 and Satb2 inhibit the expression of DCC and Unc5C, respectively, two receptors of the guidance cue Netrin-1 (*Srivatsa et al., 2014*). Since these receptors are involved in the midline crossing of corticospinal axons (*Finger et al., 2002*), one conceivable hypothesis would be that co-expression of Ctip2 and Satb2 inhibits both receptors, thus preventing subcerebral PNs to reach the spinal cord. Another explanation could be that Lmo4 inhibits neurite outgrowth by repressing the expression of the receptor tyrosine kinase Alk that plays key roles in neuritogenesis (*Lasek et al., 2011*). Alternatively, Lmo4 might interact with Lhx2, another LIM homeobox transcription factor known to regulate the expression of the guidance receptors Ephrin-A5 as well as Robo1 and 2, which in turn control different steps of corticospinal axon pathfinding (*Marcos-Mondejar et al., 2012*; *Shetty et al., 2013*). Thus, either Lmo4 expression or Ctip2 and Satb2 co-expression might control several neuronal processes allowing the correct maturation and/or refinement of PT and IT subtypes during postnatal stages of corticogenesis. We envisage that also other transcription factors normally involved in determining distinct classes of PNs during development, might take a similar postnatal function during cortical maturation.

## Conclusion

Overall, we show that area- and time-specific changes in common molecular mechanisms modulate the final features of both CPN and SCPN subpopulations in an area- and time-specific manner. More in general, our analysis provides first direct evidence of a common developmental program directing the molecular and cellular maturation of both IT and PT projection neurons. This might have important implications in the study of neocortical development, since 'serial homologies' (i.e. similar connectivity or molecular codes) among neocortical areas might be due to variations on a 'common theme', as previously suggested (*Harris and Shepherd, 2015*), rather than to multiple independent and area-specific genetic programs.

## Materials and methods

### Animals

All mouse experiments were conducted according to national and international guidelines and have been approved by the local ethical committee (CIEPAL NCE/2011-23). *Nr2f1 fl/fl* and *Lmo4 fl/fl* mice

were crossed to *Emx1-Cre* mice to inactivate either *Nr2f1* or *Lmo4* in cortical cells. WT, *Nr2f1 fl/fl*, or *Lmo4 fl/fl* were taken as controls. Mouse lines were genotyped as previously described (*Armentano et al., 2007*; *Armentano et al., 2006*; *Goebbels et al., 2006*; *Huang et al., 2009*). *Thy1-eYFP-H* mice were obtained from The Jackson Laboratory and genotyped as described in (*Feng et al., 2000*). Midday of the day of the vaginal plug was considered as embryonic day 0.5 (E0.5).

## Postmortem tissue collection

Mice at P0, P7, and P21 were intracardially perfused with paraformaldehyde (PFA) 4%. Embryonic and postnatal brain samples were fixed either for 2 hr (for immunohistochemistry) or for over-night (for *in situ* hybridization [*ISH*]) at 4°C in PFA 4%. Brain slices used for patch clamp experiments were fixed 4 hr at 4°C in 4% PFA after recording. Samples were, then, either embedded in Optimal Cutting Temperature (OCT) medium (JUNG, Germany) after being equilibrated to 30% sucrose, and cut on a Leica cryostat, or gradually dehydrated to 96% ethanol for whole-mount *ISH*. Samples to be used for floating immunofluorescences and cholera toxin-injected brains were embedded in 4% agarose after fixation and then cut on a Leica vibratome at 200 μm. No samples were excluded in this study and for each experiment at least three animals from different litters were used.

## Immunohistochemistry and immunofluorescence

Immunofluorescences on cryosections (CI) were performed as previously described (*Armentano et al., 2007*; *Armentano et al., 2006*). Floating immunofluorescence (FI) was performed on vibratome sections, which were blocked with 10% goat serum, 3% bovin serum albumin (BSA), and 0.3% triton X-100 over night at 4°C. Primary and secondary antibodies were carried out over night at 4°C. The following primary antibodies were used: mouse anti-Satb2 (dil. CI = 1:20, FI = 1:80, Abcam, UK), rat anti-Ctip2 (dil. CI = 1:300, FI = 1:400, Abcam), rabbit anti-Ctip2 (1:500, Abcam), rabbit anti-Sox5 (1:300, Gentaur, France), rabbit anti-Ski (1:50, Santa Cruz Biotechnology, Dallas, Texas), rat anti-Lmo4 (1:500, gift from Valsvader's lab), rabbit anti-Er81 (1:1000, gift from Arber's Lab), guinea pig anti-Bhlhb5 (1:50000, gift from Novitch's Lab), rabbit anti-GFP (1:1000, Molecular Probes, Eugene, Oregon), chicken anti-GFP (1:800, Abcam). The following secondary antibodies were used: goat anti-rabbit FC (488, 594, 633), goat anti-rat FC (488, 594, 633), goat anti-mouse FC (488, 594, 633), and goat anti-guinea pig FC (488, 594, 633) (dil. CI = 1:300, FI = 1:400, Life Technologies, Thermo Fisher Scientific, USA), goat anti-rabbit FC 350 (1:150, Life Technologies) and donkey anti-mouse FC 405 (1:300, Abcam). To reveal biocytin injected in patch-clamped cells, it was used Texas Red avidin D (1:500, Vector Laboratories, Burlingame, California). Slices were mounted with the following mounting solution: 80% glycerol, 2% N-propyl gallate, 1 μg/ml Hoechst (Invitrogen, Thermo Fisher Scientific, USA).

## Whole-mount *In Situ* Hybridization (*ISH*)

Whole-mount *ISH* was performed as described in (*Alfano et al., 2014*; *Alfano et al., 2011*; *Armentano et al., 2007*). The antisense *Lmo4* RNA probe was labeled using the DIG RNA labelling Kit (Roche, Switzerland) following the manufacturer's instructions.

## *pCdk5r1-Lmo4-IRES-GFP* and *pCIG2-Lmo4-IRES-GFP* plasmid construction

To synthesize the *pCdk5r1-Lmo4-IRES-GFP*, the Lmo4 ORF was amplified using available cDNA with the following primers: Mlu1-Lmo4.fw (5'GGACGCGTTGAGAGCAGCTC3') and MluI-Lmo4.rev (5'GGACGCGTTTCTGCATTACTC3'). These primers were designed with an Mlu1 restriction cassette at their 5' end. Once amplified, the Lmo4 amplicon was purified using the QIAGEN PCR Purification Kit (following manufacturer's protocols) and digested with Mlu1 (Biolabs, Ipswich, Massachusetts). The digested Lmo4 ORF was cloned into the empty *pCdk5r1-IRES-GFP* vector (digested as well with Mlu1). The plasmid was validated and sequenced using the following primer: pCDK5C.fw (5'-AGGACTAAACGCGTCGTGTCC-3').

To generate the *pCIG2-Lmo4-IRES-GFP*, the *Lmo4* variant 2 *mRNA* sequence was amplified using available P0 cDNA and the following primers: Lmo4_VAR2.FW (5'-GAAGTCCCCGAGCTGGTTTG-3') and Lmo4.REV (5'- CCATACTAGAGCAAATGTCTCTG-3'). Lmo4 amplicon was cloned into the

pCRII-TOPO vector (Invitrogen) according to manufacturer's instructions and, then, excised by using SpeI and EcoRV restriction enzymes (Biolabs). The SpeI ends was made blunt by fill-in with Klenow fragment (Biolabs). Finally, the Lmo4 fragment was cloned into a *pCIG2* plasmid (*Heng et al., 2008*) previously digested with SmaI (Biolabs). The *pCIG2 (pCAGGS-IRES-GFP2)* is a modified version of the *pCIG* vector (*Megason and McMahon, 2002*), which was obtained by inserting an IRES-GFP sequence into the *pCAGGS* vector (*Niwa et al., 1991*). Positive clones were amplified and purified by the QIAGEN Endofree Maxiprep Kit.

### *In utero* electroporation

*In utero* electroporations of *pCdk5r1-IRES-EGFP, pCdk5r1-Lmo4-IRES-EGFP, control or Lmo4-specific shRNA* were performed as previously described (*Alfano et al., 2011*; *Tabata and Nakajima, 2001*). Briefly, after a 3-cm laparotomy on deeply anesthetized pregnant females and once extroflected *uteri*, the DNA mix (1 mg/ml) was injected into the lateral ventricle of E13.5 embryos using a Femtojet microinjector (Eppendorf, Germany). The electroporations were performed on whole heads using a Tweezertrode electrode (diameter 7 mm; BTX) connected to a NEPA21 electroporator (NEPAGENE, Japan) with the following parameters: four 37 V pulses, P(on) 50 ms, P(off) 1 s, 5% decay. Then *uteri* were reallocated in the abdominal cavity, and both peritoneum and abdominal skin were sewn with surgical sutures (B. Braun Surgical, Germany).

### Transient transfection of COS cells

COS7 cells were cultured in DMEM (4.5 g/L; Invitrogen) containing 10% FCS. Transient transfections of Satb2 and Lmo4 were performed using Lipofectamine$^{TM}$ 2000 (Invitrogen) according to manufacturer's instructions at a cell confluence of 60-70%. An equal amount of *pCAG-Satb2* plasmid ( gift from V. Tarabykin's lab) was added to the cells with increasing amount of *pCIG2-Lmo4-IRES-GFP plasmid*. A *pCIG2-IRES-GFP* plasmid was used to compensate any variations in the amount of the *pCIG2-Lmo4-IRES-GFP* by maintaining the total quantity of transfected DNA constant. Cells were harvested for co-immunoprecipitation and immunoblotting 48 hr after transfection.

### Co-immunoprecipitation assayy

Co-immunoprecipitation was performed as described in (*Britanova et al., 2008*) with some modifications. Nuclear proteins were extracted from P1 control and *Nr2f1 CKO* cerebral cortices, or from harvested cells, using the NE-PER kit (Thermo Fisher Scientific) according to manufacturer's instructions. Nuclear extracts were then dialyzed against buffer D (20% glycerol, 20 mM HEPES (pH = 7.9), 100 mM KCl, 0.2 mM EDTA, 0.5 mM DTT, 0.5 mM PMSF, all from Sigma-Aldrich) in Slyde-A-Lyzer Mini Dialysis Units (3500 M.W.C.O. from Fisher Scientific, Thermo Fisher Scientific). For pre-clearing, 50 µg of the nuclear extracts were incubated with 100 µl Protein A Sepharose 50% bead slurry (Sigma-Aldrich). The pre-cleared nuclear extracts were then immunoprecipitated with either 2 µg of mouse anti-Satb2 (Abcam) or of rat anti-Lmo4 antibody (a gift from Jane Visvader's lab). A mouse anti-Brdu antibody (Sigma-Aldrich) was used as control. Immunocomplexes were collected by adding 100 µl of 50% Protein A-Sepharose bead slurry to the mix. The bound fraction was separated by pulse centrifugation and pelleted beads and input were re-suspended in 1x Nupage loading buffer (Invitrogen). Samples were loaded on a 10% SDS polyacrylamide gel and subjected to standard SDS-PAGE electrophoresis on Mini-Protean tetra cell (Biorad, Hercules, California). Then, immunocomplexes were transferred to Hybond-P membrane (Amersham, GE Healthcare, UK) via a Trans-Blot SD Semi-Dry Transfer Cell (Biorad). Immunoblotting on total nuclear extracts, bound and unbound fractions was performed with the following antibodies: rat anti-Lmo4 (1/500, gift from J. Visvader), rabbit anti-Hdac1 (1:500, Millipore, Merck, Germany), rabbit anti-Ski (1:50, Santa Cruz), mouse anti-Satb2 (1:50, Abcam), and rabbit anti-β-actin (1:500, Abcam). Primary antibodies were then detected by embedding the membrane in anti-rabbit, anti mouse or anti-rat biotinylated antibodies (1:500, Vector) and successively in ABC mix (Vector Laboratories). Revelation of the signals was performed by SuperSignal West Pico Chemiluminescent Kit (Thermo Scientific), and images were taken by Luminescent Image Analyzer LAS-3000 (Fujifilm, Japan ).

## Chromatin immunoprecipitation assay

Chromatin-immunoprecipitation (ChIP) assay on genomic DNA from controls and *Nr2f1 CKO* cortices was performed as described in (*Kuo and Allis, 1999*). Neocortices were dissected from 7 controls and *Nr2f1 CKO* P1 pups, and diced in ice cold Hanks Buffered Saline Solution. Proteins were crosslinked to DNA by adding 1% formaldehyde to the solution. The tissue was than disrupted by homogenization in lysis buffer (20 mM HEPES pH7.4, 1 mM EDTA, 150 mM NaCl, 1% SDS, 125 mM Glycine, PMSF 0.2 mg/ml). Nuclei were collected by centrifugation, re-suspended in sonication buffer (20 mM HEPES pH7.4, 1 mM EDTA, 150 mM NaCl, 0.4% SDS, PMSF 0.2mg/ml) and disrupted by 6 pulses of 10 μ amplitude in a Soniprep150 Sonicator (Sanyo). Before immunoprecipitation, samples were pre-cleared 1 hr in 50% ProteinA-Sepharose slurry and then incubated ON at 4°C with 3 μg of the following antibodies: rabbit anti-Hdac1 (Millipore), rabbit anti-H4K12 (Abcam), and a control antibody (rabbit anti-GFP, Molecular probe). An aliquot of DNA was not immunoprecipitated and used as a control; 0.5 μl of DNA from each sample were used to perform a PCR for semiquantitative analysis of the ChIP experiment using the following primers: for the *Ctip2* locus , Ctip2MAR.fw 5′-GCTTGGACTCAGTGTACCTC-3′ and Ctip2MAR.rev 5′-CAAGAAAGCACACACCGAGA-3′ and for the *Rnd2* locus, BsA.fw 5′-CGTTTGACCTTCCACCTTAG-3′ and BsA.rev: 5′-TCCCACTTGCTTGGCC-AGC-3′. PCR bands were acquired by FUJI 3000 LAS intelligent dark box equipped with a CCD camera. Hdac1 and H4k12 fold enrichment on *Ctip2* and *Rnd2* loci were tested by QPCR analysis of ChIP samples using the *LightCycler 480* Real-Time PCR System (Roche) and the above-mentioned primers.

## Retrograde labeling

For retrograde labeling, anesthetized P2 pups were placed on a stereotaxic apparatus and injected with the cholera toxin subunit B (CTB- 1 mg/ml; Invitrogen) conjugated fluorophores (Alexa Fluor, Thermo Fisher Scientific) in different brain regions. CPN in the somatosensory cortex were retrogradely labeled via injection of 92 nl of Alexa Fluor 488-conjugated CTB. Coordinates (in mm) were: AP: +1.2; and ML: 1.3 from the lambda; DV: 0.2 from the pial surface. Subcerebral injections were performed under ultrasound guidance using a Vevo 770 ultrasound backscatter microscopy system (Visual Sonics, Canada) at cervical vertebral level 1 (C1) to C2 to label corticospinal projection neurons (CSpPN), or at the midbrain-hindbrain junction to label subcerebral projection neurons (SCPN*via* 92 nl injections of Alexa Fluor 555-conjugated CTB. Dual retrograde labeling of SCPN and CPN was performed by injecting Red Retrobeads or Green IX Retrobeads (Lumafluor Inc, Durham, North Carolina), respectively, in P2 and P3 pups brains in the same regions described above. Injected pups were perfused at P7 and brains were collected as described in previous sections.

## Imaging

Images of immunostained cryosections were acquired using a DM6000 microscope (Leica, Germany) equipped with LEICA DFC 310 FX camera, while images of immunofluorescences on floating-thick sections were taken with a Zeiss 710 confocal microscope. *ISH* and whole-mount *ISH* were acquired by a Zeiss Imager Z1 microscope equipped with AXIOCAM MRm camera and a Leica Spot microscope, respectively. Images from optical and confocal microscopes were then processed using Photoshop and Zen-lite 2012 softwares, respectively.

## Counting and statistical analysis

Images of P0 coronal sections from the prospective somatosensory (pS) and the frontal/motor (F/M) regions were subdivided into 6 bins. Bins 1 and 2 represent layer VI, bins 3 and 4 represent layer V, while bins 5 and 6 represent the upper layers. At P7 the radial surface of analyzed brain regions was subdivided into 10 bins: bins 1–3 represent layer VI, bins 4–6 represent layer V, and bins 7–10 represent upper layers. Counting of single or double-labeled cells was normalized to the total number of DAPI cells in each bin. For triple immunofluorescences, the counting was performed on cortical images with a constant width of 600 μm. Each counting performed on electroporated and choleratoxin-labeled neurons was normalized to the total number of GFP- or choleratoxin-labeled cells in the layer of interest.

All the data were statistically analyzed and graphically represented using Microsoft Office Excel software. The error bars represent the standard error of the mean (SEM). Two-tailed Student's t-test was used for the analysis of statistical significance (*$p \leq 0.05$, **$p \leq 0.01$, ***$p \leq 0.001$) between to different groups.

## Morphological analysis

Neuron morphology was reconstructed in BitPlane Imaris from confocal 3D image stacks. Soma shape features were obtained using built-in Volume Rendering functionality on images with 40x magnification (criteria: surface background subtraction of 15 µm and detail smoothening of 2 µm). Extracted features of each soma shape include X, Y, and Z position of the center of mass, soma area, soma volume, oblate ellipticity, and sphericity. Calculation of these feature values was performed automatically as described in BitPlane Imaris technical documentation (http://www.bitplane.com/download/manuals/ReferenceManual6_1_0.pdf). Each soma was separated in a region of interest (ROI) of approximately 30 × 30 × 10 µm. In order to effectively render soma volume, the ROI was used to automatically compensate differences in local contrast. Limits of the soma volume were calculated automatically within ROI in most of the cases. For more detailed information, see the SI section.

Apical and basal dendrites features were obtained using Imaris Filament Tracer plugin. Features include apical dendrite diameter, total apical dendrite length (also referred as neuron radial position with respect to pial surface), number of basal dendrites, angle of each basal dendrite compared to apical one, apical dendrite ramification and bifurcation point. Apical dendrite diameter was measured at 5 µm from soma limit. Radial position and Bifurcation features were obtained from 10x image stacks. Radial position was measured as the total distance from the soma center to the pial membrane. Bifurcation was measured as the distance from the soma to the first bifurcation of the apical dendrite. Apical dendrite ramification was calculated on confocal Images of 40x as the number of secondary dendrites emerging from the apical dendrite, within a length of approximately 70 µm starting from the soma limit.

## Statistical analysis of the distinctive morphological features

Ctip2+/Satb2+ and Ctip2+/Satb2- populations were compared using non-parametric Wilcoxon Mann–Whitney U-test. Comparison of populations, based on the data collected from 40x images, was performed on n = 145 neurons. Comparison based on Radial Position and Bifurcation was performed on n = 52 neurons (52 matched neurons on 10x and 40x images). Custom clustering analysis was performed using MATLAB and to diminish scale impact on clustering, z-scale standardization was applied across all feature variables. The unsupervised k-means++ clustering algorithm was applied using squared Euclidean as distance measure. The optimal number of clusters was defined by comparing the quality of cluster separation and neuron features similarity within each cluster. The silhouette values were calculated for outcomes for each testing cycle starting from k = 2 to k = 5. After the optimal cluster number has been defined, clustering was performed with large number of iterations (iterations = 1000). MATLAB Scripts and their dependencies are available for download at https://github.com/nikiluk/signalife-moo-clust/ and could be used according to their licensing terms.

## Electrophysiology on acute slices

Whole-cell patch clamp recordings were performed on the soma of layer V YFP+ cortical neurons from 350 µm live slices of somatosensory cortices of the Thy1-eYFP-H transgenic brains. For whole-cell experiments, the slices (350 µm) were perfused with artificial cerebrospinal fluid (ACSF) which comprised (mM): 124 NaCl, 3 KCl, 26 NaHCO₃, 1.25 NaH₂PO₄, 2 CaCl₂, 1 MgSO₄, 15 D-glucose, 0.05 picrotoxin, 0.05 2-amino-5-phosphonovaleric acid (APV) and 0.02 6,7-dinitroquinoxaline-2,3-dione (DNQX), bubbled with O2:CO2: 95:5%. Visually guided, whole-cell recordings were obtained at 29°C from the soma of GFP-positive cortical neurons in layer V of somatosensory area, using patch electrodes (3–5 MΩ) filled with (in mM): 20 KCl, 100 Kgluconate, 10 HEPES, 4 Mg-ATP, 0.3 Na-GTP, 10 Na-phosphocreatine and 0.1% biocytin (Sigma Aldrich, Merck, Germany). Recordings were performed using an AxoPatch 200B amplifier (Axon Instruments, Foster City, CA), filtered at 10 kHz and were not corrected for liquid junction potentials. Data were collected using ClampEx software and analyses of recorded responses were performed using Clampfit. The cells usually had a holding

potential between -65 and -70 mV or were hold at these potentials. To calculate cell properties (resistance, time to peak, sag), currents from -200 pA to 80 pA were injected for 500 ms. The voltages were measured at the peak amplitude and at steady-state and the I-V curves were plotted. The slope of these curves corresponded to the resistance at peak (Rpeak) and at steady-state (Rss). The sag was measured as the difference of voltage at peak to the voltage at steady state when -200 pA was injected.

To measure firing and action potential properties, depolarasing currents were injected for 2 s by steps of 20 pA. The relationship between the amount of injected current and the firing frequency was then plotted. The ratio of the first interspike interval (ISI) to the last ISI was analyzed using responses recorded at two times the threshold current. To calculate action potential (AP) characteristics, we analyzed responses of cells at threshold current. Firing threshold is calculated as the interpolated membrane potential at which the derivative (dV/dt) equals 20 V/s. The firing threshold was set as the baseline to calculate characteristics of each AP (amplitude, half-width, duration, rise time). The fast afterhyperpolarization (fAHP) is the difference between the firing threshold and the minimum value found within 3 ms of the spike. The time of fAHP is the time between the peak of the spike and the fAHP. The depolarizing afterpolarization (DAP) following the fAHP (fDAP) is the difference between the maximum value obtained within 5 ms after the fAHP and the minimum value measured to calculate the fAHP. When bursts of 2 or 3 AP were fired at once, the threshold of the second AP was chosen as the maximum value to calculate the fDAP. The medium AHP (mAHP) is the difference between the threshold of AP and the minimum value found within 50 ms of the spike. The medium DAP (mDAP) is the difference between the maximum value found withnin 70 ms after the minimum value used to calculate the mAHP and this minimum value. When 2 or more APs were fired at once, the mAHP and mDAP were measured after the last AP of the burst.

## Acknowledgements

We thank J Visvader for providing us the Lmo4 antibody, HH Chen for the *shRNA Lmo4* construct, Q Lu for the *pCdk5r1-Ires-GFP*, P Arlotta for the *pCdk5r1-Fezf2 IRES-GFP* and V Tarabykin for the *pCAG-Satb2* plasmids. We are also grateful to J Sanes for giving us the permission to use the *Thy1-eYFP-H* transgenic line. We thank N Elganfoud for technical help during this project and the PRISM Microscopy Facility for technical support. J Hazan and M Nieto for constructive suggestions on the manuscript, and the whole Studer lab for fruitful discussions. This work was supported by the "Agence Nationale de la Recherche" under grant reference # ANR-13-BSV4-0011, by the French Government (National Research Agency, ANR) through the "Investments for the Future" LABEX SIGNALIFE under program reference # ANR-11-LABX-0028-01, by the "Fondation Recherche Médicale; Equipe FRM 2011" #DEQ20110421321 and by the "Fondation Jérôme Lejeune" under n° R13098AA to MS. KH was funded by a CNRS from Lebanon and AFM fellowship from France and EM was funded by an AXA Research Fund fellowship.

## Additional information

### Funding

| Funder | Grant reference number | Author |
| --- | --- | --- |
| Agence Nationale de la Recherche | ANR-13-BSV4-0011 | Michele Studer |
| Agence Nationale de la Recherche | ANR-11-LABX-0028-01 | Michele Studer |
| CNRS Lebanon | | Kawssar Harb |
| AFM-Téléthon | | Kawssar Harb |
| AXA Research Fund | | Elia Magrinelli |
| Fondation pour la Recherche Médicale | DEQ20110421321 | Michele Studer |
| Fondation Jérôme Lejeune | R13098AA | Michele Studer |

The funders had no role in study design, data collection and interpretation, or the decision to submit the work for publication.

## Author contributions

KH, MS, CA, Conception and design, Acquisition of data, Analysis and interpretation of data, Drafting or revising the article, Contributed unpublished essential data or reagents; EM, CSN, Acquisition of data, Analysis and interpretation of data, Drafting or revising the article, Contributed unpublished essential data or reagents; NL, LF, TS, DJ, Analysis and interpretation of data, Drafting or revising the article, Contributed unpublished essential data or reagents; MP, Acquisition of data, Drafting or revising the article, Contributed unpublished essential data or reagents; GS, Conception and design, Drafting or revising the article, Contributed unpublished essential data or reagents; FG, Conception and design, Analysis and interpretation of data, Drafting or revising the article

## Ethics

Animal experimentation: This study was performed in strict accordance with the recommendations in the Guide for the Care and Use of Laboratory Animals of the French Ministry of Research. All animal experiments were validated by our local ethical committees (IACUC registration number NCE2014-209).

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
