## [Decision Letter]

Thank you for submitting your work entitled "Area-specific development of distinct neocortical neuron subclasses is regulated by postnatal epigenetic modifications" for peer review at eLife. Your submission has been favorably evaluated by K VijayRaghavan (Senior editor) and three reviewers, one of whom is a member of our Board of Reviewing Editors.

The reviewers have discussed the reviews with one another and the Reviewing editor has drafted this decision to help you prepare a revised submission.

Summary:

Harb et al. describe a mechanism of postnatal epigenetic modification that regulates area-specific development of distinct projection neuron subclasses. These neurons co-express two transcription factors, Satb2 and Ctip2, that were previously thought to be mutually exclusive and to specify projections to distinct brain regions. The new work shows that the co-expressing neurons fall into two categories, projecting to the contralateral cortex or brainstem, and provides a molecular mechanism that permits co-expression. The study provides interesting data to understand the postnatal refinement of cortical projection neuron subtype diversity.

Essential revisions:

1) The in vivo evidence for a role for Lmo4 relies heavily on use of the CoupTF1 knockout in which Lmo4 is over-expressed. This is a problem because CoupTF1 knockout may have other effects on Satb2 and Ctip2 expression independent of Lmo4, including changes in a real identity. The reviewers believe it is important to provide additional in vivo evidence, e.g., from a Lmo4 transgenic or Lmo4 knockout/knockdown.

2) Additional biochemical data to support the competition model would be most helpful.

3) There is a major concern whether co-expression of Satb2 and Ctip2 is really altering neuron properties relative to neurons expressing Satb2 or Ctip2 only. The reviewers note that the co-expression only occurs post-natally, after axon targeting is already specified. Moreover, the projections of the double-positive neurons resemble those of single positive Satb2 or Ctip2 neurons, so the delayed expression of the second marker may be inconsequential. The conclusions should be modified accordingly.

4) The Lmo4-Fezf2 results do not lead to a clear molecular model and seem preliminary. The authors should add additional data or delete this section until the mechanism and significance is better understood.

5) The authors should more clearly discuss how their results relate to the effects of Ski on Satb2+ neurons.

6) The authors should be cautious in implicating the entire NuRD complex when the evidence mostly points to a role for HDAC1.

In your response letter, please also comment on the individual concerns raised by the three reviewers:-

*Reviewer #1:*

The relationship between Satb2 andCtip2 has been extensively studied. They are mostly mutually exclusive and define different classes of projection neurons. However, some studies have detected co-expressing (C/S) neurons and this paper adds a new and exciting dimension by showing that major numbers of neurons co-expressing Satb2 and Ctip2 arise after birth, particularly in S1. C/S neurons fall into subsets with distinct morphology, electrophysiology and projection patterns. C/S neurons also express Lmo4. Lmo4 deletion inhibits Ctip2 but not Satb2 expression. Over-expressing the known Ctip2 activator, Fezf2, increased Ctip2+ cells but not C/S cells, consistent with Fezf2 inhibiting Satb2. But over-expressing Lmo4 together with Fezf2 increased both Ctip2+ and C/S cells, suggesting that Lmo4 overcomes Fezf2 inhibition of Satb2. Satb2 was reported to inhibit Ctip2 via HDAC/NuRD, and Lmo4 was reported to bind HDAC1/2. The authors tested whether Lmo4 might compete with Satb2 for HDAC1. They found that co-IP of HDAC1 with Satb2 is inhibited when Lmo4 expression is increased (in CoupTF mutant cortex). This suggests a competition model where Lmo4 robs HDAC1 from the Satb2-NuRD complex on the Ctip2 promoter, allowing co-expression and the creation of new classes of neurons.

My main concern is with the biochemistry. The authors test co-IP of Satb2 and HDAC1 from control and CoupTF1 knockout, which over-expresses Lmo4. There may be other effects of CoupTF1 knockout that explain the decreased co-IP. What happens with Lmo4 CKO? There should be increased co-IP of Satb2 and HDAC1. It would also be good to blot the lysates used in Figure 7 for total Lmo4, Satb2 and HDAC1. The competition argument would also be helped by simple co-expression in tissue culture cells (e.g., HEK293). This would get around concerns that changes in co-IP could be due to changes in cellular composition of the mutant cortex.

I also wonder about the inclusion of the Lmo4-Fezf2 results at this stage. I found this part less compelling and not yet explained at the molecular level.

*Reviewer #2:*

Harb et al. described a mechanism of postnatal epigenetic modification that regulates area-specific development of distinct projection neuron subclasses.

The identity of major classes of projection neurons is established by expression of molecular determinants during cortical development. The authors aim to determine how projection neurons acquire their final properties during postnatal stages. They showed that the numbers of neurons expressing both Ctip2 and Satb2 progressively increase after birth in the somatosensory cortex. By combining retrograde tracings, the authors clearly showed that Ctip2/Satb2 co-expression defines 2 distinct neuronal subclasses projecting either to the contralateral cortex or to the brainstem. Thus Ctip2/Satb2 co-expression shapes their properties rather than determine their identity. Using Lmo4 cKO, Coup-TF1 cKO mice and IUE, they showed that Lmo4 drives this maturation program through modulation of epigenetic regulation in a time- and area-specific manner. The authors concluded that they have identified a previous unknown genetic program postnatally enhances neuronal subtype diversity in neocortical areas.

This study provides interesting data to understand the postnatal refinement of cortical projection neuron subtype diversity. Most of the data are of good quality. My biggest concern is that while the authors showed expression of Ctip2 and Satb2 goes through an Lmo4-dependent postnatal refinement, resulting in extensive co-expression in the deep-layer neurons in the somatosensory cortex, it remains unclear to me the functional significance of this co-expression. The co-expression does not define projection neuron subtype identity, as the C+S+ neurons project axons to either contralateral cortex or brainstem. Although the authors provided evidence for the existence of two subtypes of C+/S+ neurons that differ from C+/S- cells, without knowing the functional significance of Ctip2 and Satb2 co-expression, the authors' claim that the Lmo4-dependent epigenetic mechanism postnatally enhances neuronal subtype diversity in neocortical areas is stretched. My question is, how does Ctip2 and Satb2 co-expression make the neuron distinct from Ctip2+ Satb2- neurons?

I have the following concerns for the manuscript.

1) In the subsection “Morphological and electrophysiological characterization support distinct subpopulations of double Ctip2/Satb2+ neurons in S1 cortex”: To reveal the specific morphological and electrophysiological features of C+/S+ cells, the authors utilized the Thy1-eYFP-H line, and showed that most of the YFP+ cells were either C+/S- or C+/S+. However, it is as important to report among the C+/S+ deep-layer neurons or layer 5b neurons, what percentage is labeled by Thy1-eYFP-H line. If overwhelmingly the majority of the C+/S+ layer 5b cells are YFP+, using the Thy1-eYFP-H line to study the morphological and electrophysiological features of C+/S+ layer 5b cells is justified. If only a minor portion of the C+/S+ layer 5b cells are YFP+, the results obtained from using the Thy1-eYFP-H line may not be able to represent the majority of C+/S+ layer 5b neurons.

2) Discussion: "Our analysis thus identified novel subclasses of SCPNs and CPNs major neuronal groups, and unravels important roles for Ctip2, Satb2, and Lmo4 in the specification and connectivity of these subsets of PNs".

This is an overstatement. The authors did not provide evidence of how the postnatal expression of Satb2 in Ctip2+ cells, or how the expression of Ctip2 in Satb2+ cells affects the specification and connectivity of cortical PNs.

3) In the model shown in Figure 8, it is shown that the initial specification of the C+/S+ to either subcerebral or callosal occurs after birth. What is the evidence for this? It is likely that the initial specification occurs before birth, and after birth Satb2 and Ctip2 become co-expressed in some callosal neurons and in subcerebral neurons with axons targeted to the brain stem.

*Reviewer #3:*

One of the major concerns is that the authors' conclusion is not sufficiently supported by enough experimental evidence.

Experiments about the molecular mechanism shown in Figure 7 only relies on the difference between wild-type and Couptf1 cKO mice. The authors could utilize another experimental setup such as overexpression of Lmo4. In particular, in Couptf1 cKO, as the same authors previously reported, the regional identity of the somatosensory area changes into F/M-like identity. So, it is not clear if the obtained results are attributable to specific molecular mechanism (increased Lmo4 expression) or to the changes of the areal identity. Similar concern is also applicable to Figure 6, where the upregulation of Lmo4 in Couptf1 cKO was suggested to cause expansion of Ctip2+ and Ctip2/Satb2+ cells. It would be possible to examine this point by using, for example, Couptf1/Lmo4 double KO or Lmo4 knockdown.

In addition, although the authors insist the involvement of the NuRD complex in their model, presumably on the basis of the experiments about Hdac1, it does interact with many other transcriptional complexes, not only with the NuRD complex. Therefore, it is difficult to conclude the involvement of the NuRD complex, at least from the current results.

As the authors mention, Ski has been shown to regulate Satb2+ callosal neurons. However, there is a clear discrepancy, because Ski has been shown to regulate callosal neurons in a layer-dependent manner (upper layer callosal neurons, but not deep layer, are affected in the KO) (it is the opposite in this paper: less phenotype in upper layers and more in deep layers).

---

## [Author Response]

Essential revisions:

*1) The in vivo evidence for a role for Lmo4 relies heavily on use of the CoupTF1 knockout in which Lmo4 is over-expressed. This is a problem because CoupTF1 knockout may have other effects on Satb2 and Ctip2 expression independent of Lmo4, including changes in a real identity. The reviewers believe it is important to provide additional in vivo evidence, e.g., from a Lmo4 transgenic or Lmo4 knockout/knockdown.*

We appreciate the referees’ concerns about using the COUP-TFI KO as a paradigm for the mechanistic role of Lmo4 on the Ctip2 de-repression by Satb2. Unfortunately, the Lmo4 CKO model was no more available from our collaborator T. Sun, and it was not possible to obtain enough protein for a reliable co-immunoprecipitation (co-IP) after electroporating an shRNA construct against Lmo4 in wild-type cortices. Nevertheless, to overcome this issue we have followed two parallel strategies.

On one side, we aimed to demonstrate that upregulation of Lmo4 described in the COUP-TFI CKO model is directly involved in Ctip2 de-repression of layer V. To this purpose, we have downregulated Lmo4 expression in COUP-TFI mutant cortices by electroporating a shRNA construct against Lmo4. We found not only Lmo4 expression levels downregulated, as expected, but also Ctip2 expression in electroporated cells, confirming a role for Lmo4 in the increased Ctip2 expression observed in COUP-TFI CKO somatosensory areas (revised Figure 6—figure supplement 2). This result together with the in vivo evidence reported in Lmo4CKO brains (Figure 4) and Lmo4 overexpression (Figure 5), strongly support a key role for Lmo4 in modulating Ctip2 expression and allowing Ctip2 to be expressed in Satb2+ neurons.

On the other side, and as recommended by the referees, we have used an in vitro system to support the de-repression model (see below for more details).

*2) Additional biochemical data to support the competition model would be most helpful.*

To support the competition model, we performed Co-immunoprecipitaion (Co-IP) experiments on protein extracts from COS7 cells. By progressively increasing the amount of Lmo4-expressing vector transfected in cells and maintaining the amount of co-transfected Satb2-expressing vector constant, we observed a progressive increase in the interaction between Hdac1 and Lmo4 at the expense of Satb2. These new data presented in Figure 7 strongly support previous Co-IP results obtained on protein extracts from COUP-TFI mutant cortices (in Figure 7).

*3) There is a major concern whether co-expression of Satb2 and Ctip2 is really altering neuron properties relative to neurons expressing Satb2 or Ctip2 only. The reviewers note that the co-expression only occurs post-natally, after axon targeting is already specified. Moreover, the projections of the double-positive neurons resemble those of single positive Satb2 or Ctip2 neurons, so the delayed expression of the second marker may be inconsequential. The conclusions should be modified accordingly.*

The fact that Ctip2 and Satb2 are co-expressed post-natally is to our opinion in accordance with the maturation process of projection neurons. We do not want to claim that the co-expression of these factors specifies a complete new class of neurons, but instead that this particular molecular code is involved in the final refinement/maturation of cortical PN major classes leading to subtype diversity. It is therefore not surprising that double-expressing neurons project to similar targets as single-expressing neurons (callosal vs. subcerebral) since we do not claim that the post-natal co-expression of Ctip2 and Satb2 would drastically change the phenotype of these neurons. We instead propose that other cellular properties normally acquired after birth might be under the control of different molecular codes. These properties could include dendritic maturation, axonal remodeling and acquisition of collateral projections, modification of neuronal excitability and/or others. Since neurons expressing Ctip2 and Satb2 differ molecularly, morphologically and electrophysiologically from single Ctip2+ neurons, we consider them as distinct subtypes of major classes.

We modified our conclusions accordingly and better clarified our model (see also new Figure 8).

Finally, we strongly believe that our work goes beyond the identification and characterization of novel transcriptional determinants involved in specifying the identity of distinct PN classes, and opens a novel prospective in deciphering molecular mechanisms required in post-natal neuronal refinement. We speculate that beside Ctip2 and Satb2, other transcriptional regulators known to be mutually exclusive during development, might be co-expressed after birth in several regions of the nervous system. The overall significance of this co-expression will be certainly the topic of future studies.

*4) The Lmo4-Fezf2 results do not lead to a clear molecular model and seem preliminary. The authors should add additional data or delete this section until the mechanism and significance is better understood.*

We initially added these data to explain the lack of Ctip2 activation in LMO4-expressing upper layer neurons. But we agree that the data are at this stage preliminary and have therefore deleted this section as requested.

*5) The authors should more clearly discuss how their results relate to the effects of Ski on Satb2+ neurons.*

In this work, we do not intend to state that Ski is involved in the de-repression mechanism leading to the specification of double Ctip2/Satb2-expressing neurons. On the contrary, we show that despite maintenance of high Ski expression post-natally, double C/S+ cells are increasingly generated from P0 to P21, implying that Ski is not involved in Ctip2 de-repression observed after birth. We instead show that Lmo4 disrupts the repression mechanism independently of the presence of Ski in the NuRD complex. To fully support this last statement, we have now added a new experiment by co-immunoprecipitating control and COUP-TFI CKO cortical protein extracts with Lmo4 and assessed its interaction with Ski and Satb2 proteins by Western blot. As shown in the new panel B of Figure 7, no changes in Lmo4-Ski nor Lmo4-Satb2 interactions have been detected between controls and CKOs, indicating that Lmo4 acts on Ctip2 de-repression by binding Hdac1 independently of Ski and/or Satb2.

*6) The authors should be cautious in implicating the entire NuRD complex when the evidence mostly points to a role for HDAC1.*

We agree with the reviewers and along the text we have put less emphasis on the implication of the entire NuRD complex in the de-repression mechanism. Nevertheless, the aim of our work is to specifically study Hdac1 in the context of the Ctip2 locus in which the NuRD complex is actively implicated. Since interaction between Hdac1 and NuRD complex seems to be the major mechanism by which Satb2 inhibits Ctip2 expression, we assumed that high levels of Lmo4 would specifically act in this context. Finally, we refer to the NuRD complex because Satb2 interacts with Hdac1 through this complex and not directly. Thus, Hdac1 sequestration by Lmo4 limits interactions between Satb2-NuRD complex and Hdac1.

Reviewer #1:

*My main concern is with the biochemistry. The authors test co-IP of Satb2 and HDAC1 from control and CoupTF1 knockout, which over-expresses Lmo4. There may be other effects of CoupTF1 knockout that explain the decreased co-IP.*

We agree with the reviewer that the use of the COUP-TFI CKO might not be ideal to directly proof the biochemical mechanisms leading to the Ctip2/Satb2 co-localization. We have now repeated the experiment in an in vitro system, as requested, and thus in a COUP-TFI-independent environment (see more below).

*What happens with Lmo4 CKO? There should be increased co-IP of Satb2 and HDAC1.*

Unfortunately we could not perform this experiment in the cortex of Lmo4 CKOs, since this line is not kept in our animal facility but maintained as a very small colony in NY by our collaborator Tao Sun who was unable to send us pregnant females for performing co-IP experiments. To overcome this problem, we have electroporated a shRNA construct against Lmo4 in the COUP-TFI mutant brain (as also requested by reviewer 3) and demonstrated that decreased Lmo4 is sufficient to rescue the increased Ctip2 expression in layer V (now in Figure 6—figure supplement 2). This implies a positive correlation between Lmo4 and Ctip2 expression.

*It would also be good to blot the lysates used in Figure 7 for total Lmo4, Satb2 and HDAC1.*

We have now presented the requested controls that were already performed at the time of the experiments. The Western blot is now in Figure 7.

*The competition argument would also be helped by simple co-expression in tissue culture cells (e.g., HEK293). This would get around concerns that changes in co-IP could be due to changes in cellular composition of the mutant cortex.*

We thank the reviewer for this helpful suggestion and have used COS7 cells as our reliable in vitro system, to support the competition model. To test if increasing levels of Lmo4 may interfere with Satb2-Hdac1 interaction we co-transfected increasing amounts of Lmo4-expressing vector with equal amounts of Satb2-expressing one. As expected, we observed the lowest levels of Hdac1-Satb2 interaction in correspondence to the highest amounts of Lmo4 expression. These results strongly support previous Co-IP experiments performed on COUP-TFI mutant cortices extracts.

*I also wonder about the inclusion of the Lmo4-Fezf2 results at this stage. I found this part less compelling and not yet explained at the molecular level.*

We eliminated the part dealing with the co-expression of Lmo4 and Fezf2 and amended the figure as appropriate. Figure 5 shows now a statistically significant increase of single Ctip2+ and double C/S+ cells in Lmo4 overexpressing layer V neurons when compared to control GFP electroporated cells.

Reviewer #2:

*This study provides interesting data to understand the postnatal refinement of cortical projection neuron subtype diversity. Most of the data are of good quality. My biggest concern is that while the authors showed expression of Ctip2 and Satb2 goes through an Lmo4-dependent postnatal refinement, resulting in extensive co-expression in the deep-layer neurons in the somatosensory cortex, it remains unclear to me the functional significance of this co-expression. The co-expression does not define projection neuron subtype identity, as the C+S+ neurons project axons to either contralateral cortex or brainstem. Although the authors provided evidence for the existence of two subtypes of C+/S+ neurons that differ from C+/S- cells, without knowing the functional significance of Ctip2 and Satb2 co-expression, the authors' claim that the Lmo4-dependent epigenetic mechanism postnatally enhances neuronal subtype diversity in neocortical areas is stretched. My question is, how does Ctip2 and Satb2 co-expression make the neuron distinct from Ctip2+ Satb2- neurons?*

While we agree with the reviewer that we still do not fully understand the functional significance of the Ctip2 and Satb2 co-expression in layer V neurons, the aim of this work was mainly to demonstrate that mutually exclusive transcription factors known to determine the identity of distinct projection neuron classes during development, can again co-localize post-natally by, most probably, determining cellular properties that normally neurons acquire after birth. These properties could include dendritic maturation, axonal remodelling, acquisition of collateral projections, refinement of neuronal excitability and/or others. Since neurons expressing Ctip2 and Satb2 differ molecularly, morphologically and electrophysiologically from single Ctip2+ neurons, we consider them as subtypes of major classes. In this sense, we meant an increase of neuronal diversity, since further subclasses will be specified after birth. In addition, even if not the unique player, our data show that Lmo4 contributes in this process. We have nevertheless put less emphasis along the text on the importance of Ctip2 and Satb2 in enhancing neuronal subtype diversity in neocortical areas, as requested by the referee.

*I have the following concerns for the manuscript.*

*1) In the subsection “Morphological and electrophysiological characterization support distinct subpopulations of double Ctip2/Satb2+ neurons in S1 cortex”: To reveal the specific morphological and electrophysiological features of C+/S+ cells, the authors utilized the Thy1-eYFP-H line, and showed that most of the YFP+ cells were either C+/S- or C+/S+. However, it is as important to report among the C+/S+ deep-layer neurons or layer 5b neurons, what percentage is labeled by Thy1-eYFP-H line. If overwhelmingly the majority of the C+/S+ layer 5b cells are YFP+, using the Thy1-eYFP-H line to study the morphological and electrophysiological features of C+/S+ layer 5b cells is justified. If only a minor portion of the C+/S+ layer 5b cells are YFP+, the results obtained from using the Thy1-eYFP-H line may not be able to represent the majority of C+/S+ layer 5b neurons.*

We agree with the reviewer and quantified the percentage of double C/S+ deep-layer neurons labeled by the Thy1-eYFP-H mouse line. We now report in the text that 56% of double C/S+ cells express YFP. Thus, together with the previous quantification that 77% of Thy1-eYFP-expressing cells are double positive for Ctip2 and Satb2, we believe that the Thy1-eYFP-H line represents an appropriate tool to undertake a morphological and electrophysiological analysis of these cells.

*2) Discussion: "Our analysis thus identified novel subclasses of SCPNs and CPNs major neuronal groups, and unravels important roles for Ctip2, Satb2, and Lmo4 in the specification and connectivity of these subsets of PNs".*

*This is an overstatement. The authors did not provide evidence of how the postnatal expression of Satb2 in Ctip2+ cells, or how the expression of Ctip2 in Satb2+ cells affects the specification and connectivity of cortical PNs.*

We agree with the reviewer that we did not directly prove that forced ectopic expression of Satb2 into Ctip2+ cells or Ctip2 into Satb2+ cells at post-natal stages can alter the molecular, morphological, electrophysiological and connectivity properties of single-expressing cells. Because of the amount of work, we think that this part would be the topic of a further study. Thus, we agree to tone down our assessment and amended the sentence as follows “Our analysis thus identified novel subclasses of SCPNs and CPNs major neuronal groups with distinct features, which can be specifically identified by Ctip2/Satb2 co-localization.”

*3) In the model shown in Figure 8, it is shown that the initial specification of the C+/S+ to either subcerebral or callosal occurs after birth. What is the evidence for this? It is likely that the initial specification occurs before birth, and after birth Satb2 and Ctip2 become co-expressed in some callosal neurons and in subcerebral neurons with axons targeted to the brain stem.*

We fully agree that we do not know the exact time when the double C/S+ subpopulation is established, but only when it becomes recognizable at the protein level. We have now amended the model and eliminated the P0-P7 timing.

Moreover, we agree that we showed no causative relations between Ctip2/Satb2 co-expression and connectivity refinement. Our scheme was meant only as a hypothetical model resuming what is known up to date from our present work and related work published in literature. We have tried to clarify these issues in the new model presented in Figure 8.

Reviewer #3:

*One of the major concerns is that the authors' conclusion is not sufficiently supported by enough experimental evidence. Experiments about the molecular mechanism shown in Figure 7 only relies on the difference between wild-type and Couptf1 cKO mice. The authors could utilize another experimental setup such as overexpression of Lmo4. In particular, in Couptf1 cKO, as the same authors previously reported, the regional identity of the somatosensory area changes into F/M-like identity. So, it is not clear if the obtained results are attributable to specific molecular mechanism (increased Lmo4 expression) or to the changes of the areal identity. Similar concern is also applicable to Figure 6, where the upregulation of Lmo4 in Couptf1 cKO was suggested to cause expansion of Ctip2+ and Ctip2/Satb2+ cells. It would be possible to examine this point by using, for example, Couptf1/Lmo4 double KO or Lmo4 knockdown.*

We fully agree with the reviewer and to respond to her/his concerns, we have set up a new series of electroporation experiments by inhibiting Lmo4 expression via an Lmo4-specific shRNA in COUP-TFI CKO cortices. In the revised Figure 6—figure supplement 2, we first show that the electroporation of a control shRNA does not affect the radial expression of Lmo4 (Panel A) and Ctip2 (panel B), and that electroporated cells show the migratory defect previously described in COUP-TFI mutant brains (Alfano et al., Development, 2011). On the contrary, electroporating mutant brains with the Lmo4-specific shRNA drastically decreases the abnormally expanded Ctip2 radial expression in the upper part of layer V (panel D). Thus, these data strongly suggest that the abnormal Ctip2 expression in layer V of COUP-TFI CKO cortices is primarily due to the high abnormal levels of Lmo4 expression observed in COUP-TFI CKO brains.

*In addition, although the authors insist the involvement of the NuRD complex in their model, presumably on the basis of the experiments about Hdac1, it does interact with many other transcriptional complexes, not only with the NuRD complex. Therefore, it is difficult to conclude the involvement of the NuRD complex, at least from the current results.*

The referee is perfectly correct by stating that Hdac1 interacts with many other transcriptional complexes beside the NuRD complex; however, the aim of our work is to specifically study Hdac1 in the context of the Ctip2 locus in which Satb2 interacts with this histone deacetylase through the NuRD complex. Since interaction between Hdac1 and NuRD complex has been widely reported in the literature (Alcamo et al., 2008; Britanova et al., 2008; Baranek et al., 2012) and seems to be the major mechanism by which Satb2 inhibits Ctip2 expression, we assumed Lmo4 acts by sequestering Hdac1 before its interaction with the NuRD complex, which is assembled on the Ctip2 locus through interaction with Satb2. In accordance with the referee, we have now put less emphasis on the implication of the entire NuRD complex in the de-repression mechanism.

*As the authors mention, Ski has been shown to regulate Satb2+ callosal neurons. However, there is a clear discrepancy, because Ski has been shown to regulate callosal neurons in a layer-dependent manner (upper layer callosal neurons, but not deep layer, are affected in the KO) (it is the opposite in this paper: less phenotype in upper layers and more in deep layers).*

We fully agree with the reviewer and indeed we never state that Ski is involved in the specification mechanism of double Ctip2/Satb2-expressing cells. On the contrary, we show in Figure 4—figure supplement 1 that despite high levels of Ski expression maintained after birth, the number of double C/S+ are increasingly generated and several are also positive for Ski, implying that the Ctip2 de-repression is Ski-independent. This is different from what normally happens during embryonic stages of development in which Ski interacts with Satb2 in the Ctip2 repression mechanism (as reported in Baranek et al., PNAS, 2011). Our study instead shows that Lmo4 disrupts the repression mechanism independently of the Ski-mediated mechanism. To fully support this last statement, we have now added a new Co-IP experiment by co-immunoprecipitating control and COUP-TFI CKO cortical protein extracts with the Lmo4-specific antibody and assessed the amount of bound Ski and Satb2 proteins by Western blot. As seen in the new version of Figure 7, no changes in protein interactions have been detected between the two conditions, indicating that Lmo4-mediated de-repression of Ctip2 does not involve Ski and/or Satb2.